# Osteology of an exceptionally well-preserved tapejarid skeleton from Brazil: Revealing the anatomy of a curious pterodactyloid clade

**Victor Beccari**[1,2,3]*, **Felipe Lima Pinheiro**[4], **Ivan Nunes**[5], **Luiz Eduardo Anelli**[6], **Octávio Mateus**[2,3], **Fabiana Rodrigues Costa**[7]

**1** Instituto de Biociências, Universidade de São Paulo, São Paulo, São Paulo, Brazil, **2** Faculdade de Ciências e Tecnologia, FCT, GeoBioTec, Department of Earth Sciences, Universidade Nova de Lisboa, Caparica, Portugal, **3** Museu da Lourinhã, Lourinhã, Portugal, **4** Laboratório de Paleobiologia, Universidade Federal do Pampa, São Gabriel, Rio Grande do Sul, Brazil, **5** Instituto de Biociências, Laboratório de Herpetologia (LHERP), Universidade Estadual Paulista, São Vicente, São Paulo, Brazil, **6** Instituto de Geociências, Universidade de São Paulo, Cidade Universitária, São Paulo, São Paulo, Brazil, **7** Centro de Ciências Naturais e Humanas, Laboratório de Paleontologia de Vertebrados e Comportamento Animal (LAPC), Universidade Federal do ABC, São Paulo, São Paulo, Brazil

* victor.beccari@gmail.com

**Data Availability Statement:** The Supplemental Files include the dataset adopted for the phylogenetic analysis. The specimen described in this study is stored in the collection of the Coleção

## Abstract

A remarkably well-preserved, almost complete and articulated new specimen (GP/2E 9266) of *Tupandactylus navigans* is here described for the Early Cretaceous Crato Formation of Brazil. The new specimen comprises an almost complete skeleton, preserving both the skull and post-cranium, associated with remarkable preservation of soft tissues, which makes it the most complete tapejarid known thus far. CT-Scanning was performed to allow the assessment of bones still covered by sediment. The specimen can be assigned to *Tupa. navigans* due to its vertical supra-premaxillary bony process and short and rounded parietal crest. It also bears the largest dentary crest among tapejarine pterosaurs and a notarium, which is absent in other representatives of the clade. The new specimen is here regarded as an adult individual. This is the first time that postcranial remains of *Tupa. navigans* are described, being also an unprecedented record of an articulated tapejarid skeleton from the Araripe Basin.

## Introduction

The pterosaur clade Tapejaridae was a major component of Early Cretaceous continental faunas, achieving a widespread distribution in Gondwana and Eurasia (e.g. [1–3]). Tapejarids are characterized by their edentulous jaws and often huge cranial crests [1,2], and are sometimes inferred to have had an herbivorous diet [4]. In Brazil, tapejarids are among the most abundant and diverse pterosaur taxa, recovered from the Crato and Romualdo *Lagerstätten* (Araripe Basin, northeastern part of the country) and from the desertic environments of the Goiô-Erê Formation (Paraná Basin, southern Brazil) [5]. Most Brazilian tapejarids are known from isolate skulls or partial skeletons, with the exceptions of *Caiuajara dobruskii* and *Tapejara*

de Paleontologia Sistemática of the Geosciences Institute of Universidade de São Paulo under reference number: GP/2E 9266. Data inquiries may be sent to the data curator Juliana de Moraes Leme at leme@usp.br. The raw 3D data for the specimen can be found in the online repository of MorphoSource at: ark:/87602/m4/369632.

**Funding:** FLP is supported by grants from Conselho Nacional de Desenvolvimento Científico e Tecnológico (CNPq process numbers 407969/ 2016-0, 305758/2017-9) and Fundacão de Amparo à Pesquisa do Estado do Rio Grande do Sul (FAPERGS process number 16/2551-0000271-1). OM is supported by grants from GeoBioTec-GeoBioSciences, GeoTechnologies and GeoEngineering NOVA [GeoBioCiências, GeoTecnologias e GeoEngenharias], grant UIDB/ 04035/2020 by the Fundação para a Ciência e Tecnologia. FRC is supported by Conselho Nacional de Desenvolvimento Científico e Tecnológico (CNPq) for support (grant No. 421772/2018-2).

**Competing interests:** The authors have declared that no competing interests exist.

*wellnhoferi*, from which several disarticulated specimens were recovered [5,6]. Up to now, the most complete tapejarid specimens were found in the Early Cretaceous of China [7], but their anatomy has not yet been described in detail.

The genus *Tupandactylus*, perhaps the most impressive tapejarid known, due to its large soft-tissue sagittal crest, is comparatively abundant in Crato Formation limestones, with several specimens deposited in public and private collections (e.g. [8–10]). However, both *Tupandactylus* species—*Tupa. imperator* [8] and *Tupa. navigans* [9]—are so far known solely from isolated skulls [10].

Because the typical Crato Formation preservation hinders a complete preparation and isolation of bones, most pterosaur specimens from this unit were described using solely their superficially exposed features. Here, we describe a nearly complete, almost fully articulated *Tupa. navigans* skeleton (GP/2E 9266) with aid of CT-Scanning. Apart from presenting the first postcranial material unambiguously assigned to *Tupandactylus*, the new specimen is indeed the best-preserved tapejarid skeleton known so far, shedding new light on the anatomy of this pterodactyloid clade.

## Geological setting

Specimen GP/2E 9266 is preserved in six perfectly complementary yellowish limestone slabs, which fit together by rectilinear cuts. The cutting pattern reflects a typical procedure of quarryman to extract paving stones from Crato Formation outcrops. Although the exact locality and horizon from where GP/2E 9266 was recovered are unknown, we are sure of its provenance from Crato Formation because the lithology of the embedding matrix perfectly fits the biomicritic laminated limestone beds of that stratigraphic unit. Among the sedimentary deposits of the Araripe Basin (northeastern Brazil), the Crato Formation rivals the younger Romualdo Formation in abundance and exceptional preservation of their fossils. Regarded as Aptian in age, the Crato Formation crops out following a mainly N-SE belt in the northern scarps of the Araripe plateau [11]. The genesis of Crato Formation laminated limestones is presumably related to authigenic carbonate precipitation and deposition following seasonal phytoplankton blooms or seasonal salinity fluctuations caused by evaporation [12,13]. Crato Formation carbonates were deposited in a quiet and protected environment, with evidence of a strong chemocline, especially concerning salinity and oxygen concentration. The abundance of freshwater parautochthonous fauna (as Ephemeroptera larvae and anurans), in association to halite pseudomorphs, indicates fresh shallow waters above an at least episodic hypersaline bottom. Similarly, the absence of benthic fauna and bioturbated sediments indicate that deep waters were anoxic [12–14].

## Materials and methods

### Material

The new specimen is permanently housed at Laboratório de Paleontologia Sistemática of the Instituto de Geociências at Universidade de São Paulo (São Paulo, Brazil) under the collection number GP/2E 9266. The specimen was intercepted during a police raid at Santos Harbour, São Paulo State, Brazil, and confiscated together with several other exceptionally well-preserved fossils, which are now fully accessible for research. No permits were required for the described study, which complied with all relevant regulations. It is preserved in six limestone slabs, four large square-cut plates comprising most of the skeletal elements and soft-tissue crest (slabs 1 to 4, from upper left to bottom right), and two smaller rectangular-cut ones (slabs 5 and 6, bottom left to right). When joint together these slabs perfectly tie all parts and bones that had their pieces separated by these cuts. The specimen presents an exquisite

**Table 1. *Tupandactylus navigans* GP/2E 9266 preserved elements with comments on the preservation.**

| Bone(s) | State of preservation | Comments |
|---|---|---|
| Cranial bones | Varies, mainly laterally compressed | Most bones preserved; premaxillomaxilla divided in slabs 1 and 3; parietal crest fragmented |
| Mandible | Laterally compressed | Right mandibular ramus ventrally deflected |
| Hyoids | Preserved with minor distortion | Three ceratobranchials |
| Non-ossified tissue | Impression | Rhamphotheca present; sagittal crest divided into slabs 1, 2 and 3; dorsalmost margin of sagittal crest missing |
| Cervical vertebrae | Preserved with minor distortion | Nine cervical vertebrae preserved; atlantoaxis fused; cervical vertebrae 8 and 9 divided into slabs 3 and 4 |
| Notarium | Laterally compressed | Composed by five dorsal vertebrae; left surface weathered |
| Dorsal vertebrae | Laterally compressed | Five free dorsal vertebrae; left surface weathered |
| Sacral vertebrae | Laterally compressed | Five fused sacral vertebrae; greatly weathered |
| Caudal vertebrae | Preserved with minor distortion | Five caudal vertebrae preserved |
| Dorsal ribs | Preserved with minor distortion | Nine dorsal ribs preserved; mainly fragmented |
| Scapulocoracoid (l) | Preserved with minor distortion | - |
| Scapulocoracoid (r) | Preserved with minor distortion | Divided into slabs 3 and 4 |
| Sternum | Dorsoventrally compressed | Divided into slabs 3 and 4 |
| Sacrum (l) | Preserved with minor distortion | Missing prepubis |
| Humerus (l) | Dorsoventrally compressed | Divided into slabs 4 and 6; highly flattened |
| Humerus (r) | Preserved with minor distortion | - |
| Ulna (l) | Dorsoventrally compressed | Divided into slabs 4 and 6; highly flattened |
| Ulna (r) | Dorsoventrally compressed | Proximally retains its form |
| Radius (l) | Dorsoventrally compressed | Divided into slabs 4 and 6; highly flattened |
| Radius (r) | Dorsoventrally compressed | Flattened |
| Carpals (l) | Preserved with minor distortion | 3 carpal elements preserved |
| Carpals (r) | Laterally compressed | 3 carpal elements and pteroid preserved |
| Metacarpals I-III (l) | Preserved with minor distortion | Distally articulated |
| Metacarpals I-III (r) | Preserved with minor distortion | Proximally articulated |
| Metacarpal IV (l) | Dorsoventrally compressed | Missing distal articulation surface |
| Metacarpal IV (r) | Dorsoventrally compressed | Divided into slabs 3 and 4 |
| Manus (l) | Preserved with minor distortion | Articulated elements; complete |
| Manus (r) | Preserved with minor distortion | Disarticulated elements, divided into slabs 3 and 4 |
| Wing phalanxes (l) | Preserved with minor distortion | Articulated elements; divided into slabs 3, 4 and 5; missing fourth wing phalanx |
| Wing phalanxes (r) | Preserved with minor distortion | Articulated elements; minor fractures |
| Femur (l) | Laterally compressed | Missing femoral head; distally retains its form |
| Femur (r) | Preserved with minor distortion | - |
| Tibia and fibula (l) | Laterally compressed | - |
| Tibia and fibula (r) | Laterally compressed | Distally retains its form |
| Tarsals (l) | Preserved with minor distortion | Three elements preserved |
| Tarsals (r) | Preserved with minor distortion | Three elements preserved |
| Metatarsals (l) | Preserved with minor distortion | Four elements preserved; fifth metatarsal absent, divided into slabs 4 and 6 |
| Metatarsals (r) | Preserved with minor distortion | Four elements preserved; fifth metatarsal absent |
| Pes (l) | Preserved with minor distortion | Articulated elements; complete |
| Pes (r) | Preserved with minor distortion | Disarticulated elements; missing distal phalanx and unguals |

preservation of soft-tissue elements, and most of the preparation was done before its incorporation to the collection. Both left square-cut slabs (1 and 3) have been broken and were then rejoined with thin metal bars, also prior to the incorporation. The preserved skeletal elements show different degrees of taphonomic distortion (Table 1).

## Phylogenetic analysis

The holotype of *Tupa. navigans* SMNK PAL 2344 was initially coded as a separate OUT, but no character differed from the scoring of GP/2E 9266. Therefore, the phylogenetic position of *Tupa. navigans* was accessed through the scoring of GP/2E 9266 using the character-taxon matrix of [15], composed by 64 taxa (including the new specimen) and 150 discrete characters (supplementary data 1). It is worth mentioning that the dataset used derived from previous studies (i.e., [2,16,17]) and in those analysis, *Tupa. navigans* was not scored. Parsimony analyses were performed using TNT v. 1.5 (*Tree Analysis Using New Technology* [18]) Traditional Search algorithm, with Wagner trees builds followed by tree bisection reconnection (TBR), and branch swapping with a hold of 20 and 1,000 replicates, random seed and collapsing trees after the search. A second analysis using the same dataset and parameters was conducted including the tapejarid specimen MN 6588-V [19], a post-cranial material, as a new OTU, to assess its relationships with other tapejarids (supplementary data 2). The consensus tree was recovered using the "Consensus > Strict" option of TNT v. 1.5.

## Computed Tomography (CT) scanning

The X-ray CT-Scanning was made at Hospital Universitário, Universidade de São Paulo (São Paulo, Brazil), using a Philipp Brilliance 64 medical tomograph. The voxel size of the data is 0.976 mm, with an overlap of 0.33 mm. The resulting tomographic slices were processed and segmented using AVIZO 9.2. The raw 3D data is available at MorphoSource (ark:/87602/m4/369632). The scan stack was upscaled to twice its initial volume using the Resample module and segmented with the brush tool. The generated meshes were smoothed in Blender 2.9 using Laplacian Smooth and Remesh modifiers, and the rendered images were treated in Adobe Photoshop CC 21.1.1.

## Institutional abbreviations

AMNH, American Museum of Natural History (New York, USA); CAD, CPCA, Centro de Pesquisas Paleontológicas da Chapada do Araripe (Departamento Nacional de Produção Mineral, Crato, Brazil); GP/2E, Laboratório de Paleontologia Sistemática do Instituto de Geociências da Universidade de São Paulo (São Paulo, Brazil); IMCF, Iwaki Coal and Fossil Museum (Iwaki, Japan); MCT, Museu de Ciências da Terra (Departamento Nacional de Produção Mineral, Rio de Janeiro, Brazil); MN, Museu Nacional (Rio de Janeiro, Brazil); MPSC, Museu de Paleontologia (Santana do Cariri, Brazil); NSM, National Science Museum (Tokyo, Japan); SMNK, Staatliches Museum für Naturkunde (Karlsruhe, Germany); YPM, Yale Peabody Museum of Natural History (New Haven, USA); ZIN, Zoological Institute of the Russian Academy of Sciences (Saint Petersburg, Russia).

## Results

### Systematic palaeontology

Pterosauria Kaup, 1834 [20]
Pterodactyloidea Plieninger, 1901 [21]
Tapejaridae Kellner, 1989 [22]
Tapejarinae Kellner & Campos, 2007 [1]
*Tupandactylus* Kellner & Campos, 2007 [1]
*Tupandactylus navigans* (Frey, Martill & Buchy, 2003 [9])

   **Material.**   GP/2E 9266, an almost complete skeleton with associated soft tissue remains (Table 1; Fig 1).

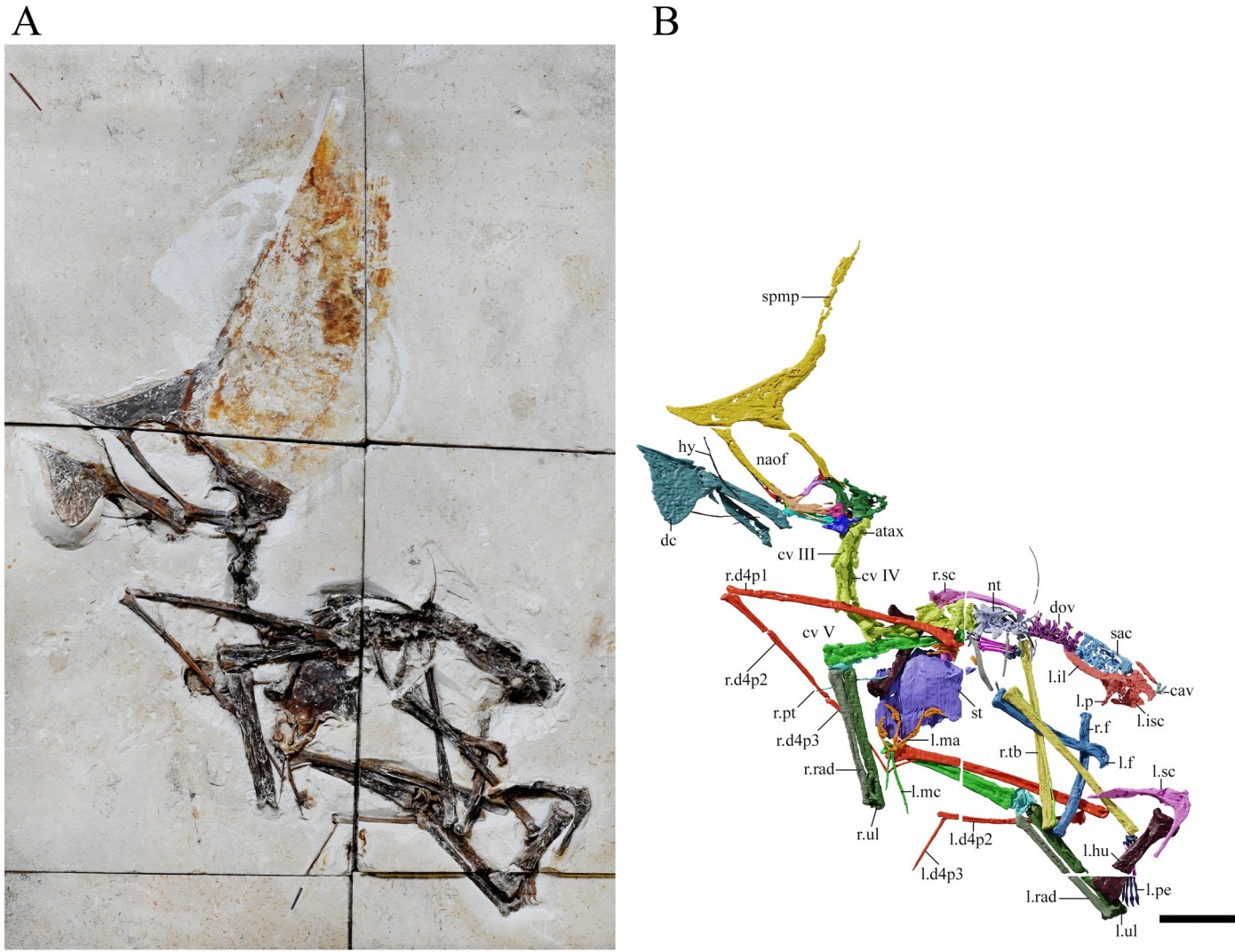

**Fig 1. *Tupandactylus navigans* GP/2E 9266.** Photo of specimen (A); 3D model of specimen (B). Abbreviations: atax, atlas-axis complex; cav, caudal vertebrae; cv, cervical vertebrae; d4, digit four; dc, dentary crest; dov, dorsal vertebrae; f, femur; hu, humerus; il, ilium; isc, ischium; ma, manus; mc, metacarpal; naof, nasoantorbital fenestra; not, notarium; p, pubis; pe, pes; pmc, premaxillary crest; pt, pteroid; rad, radius; sac, sacral vertebrae; sc, scapulocoracoid; spmp, supra-premaxilar bony process; st, sternum; tar, tarsals; tf, tibiofibula; ul, ulna. Scale bar = 50 mm.

**Horizon and locality.** Crato Formation, Santana Group, Araripe Basin, NE-Brazil. Early Cretaceous (Albian). Exact locality undetermined.

**Revised diagnosis.** *Tupandactylus navigans* can be distinguished from other tapejarid pterosaurs by 1) previously defined autapomorphies [9]: premaxillomaxilla concave anteriorly; a striated premaxillary crest; supra-premaxillary bony process perpendicular to the long axis of the skull; parietal crest short and rounded; 2) autapomorphies identified in GP/2E 9266: anteriorly deflected expansion of premaxillary crest; deep and blade-shaped dentary crest with subvertical posterior margin; lateral surfaces of cervical vertebrae postzygapophyses with longitudinal grooves.

**Table 2.** *Tupandactylus navigans* GP/2E 9266 cranium measurements.

| Bone | Comments | Measurement (mm) |
|---|---|---|
| Cranium | Length from tip of premaxilla to squamosal | 286.7 |
| Cranium | Height at the quadrates | 88.9 |
| Cranium | Maximum height from dorsalmost tip of sagittal crest to ventral margin of the skull | 522.3 |
| Rostrum | Length from tip of premaxilla to anterior of nasoantorbital fenestra | 74.4 |
| Rostrum | Height of the anteriormost point of the nasoantorbital fenestra | 28.9 |
| Rostrum | Height anterior to nasoantorbital fenestra | 161.1 |
| Rostrum | Ventral deflection angle relative to ventral margin of the skull | 151˚ |
| Rostral Value | RV ratio *sensu* Kellner 2010; 2017 | 2.57 |
| Rostral Index | RI ratio *sensu* Martill & Naish, 2006 | 0.46 |
| Premaxillomaxilla | Length from tip to jugal | 207.9 |
| Supra-premaxillary process | Preserved height | 137.4 |
| Sagittal crest | Anterior margin height from dorsal tip of premaxillary crest | 342.9 |
| Sagittal crest | Maximum height from dorsalmost tip of sagittal crest to dorsal margin of the skull | 470.9 |
| Nasoantorbital fenestra | Anteroposterior length | 129.8 |
| Nasoantorbital fenestra | Dorsoventral height | 64.6 |
| Jugal (l) | Maxillary process length | 40.5 |
| Jugal (l) | Lacrimal process length | 31.8 |
| Jugal (l) | Postorbital process length | 51.0 |
| Jugal (l) | Inclination angle of lacrimal process relative to ventral margin of the skull | 101˚ |
| Jugal (l) | Inclination angle of postorbital process relative to ventral margin of the skull | 140˚ |
| Jugal (l) | Inclination angle of quadrate process relative to ventral margin of the skull | 151˚ |
| Quadrate (l) | Inclination angle relative to ventral margin of the skull | 148˚ |
| Mandible | Length from tip of dentary to retroarticular process | 229.6 |
| Mandible | Ventral deflection angle relative to dorsal margin of the mandible | 168˚ |
| Symphysis | Length from tip of dentary to the posterior margin of the symphysis | 94.1 |
| Mandible | Rami separation angle | 20˚ |
| Mandible | Mid-shaft height | 16.4 |
| Mandible | Mid-shaft width | 4.1 |
| Dentary Crest | Anteroposterior length | 103.3 |
| Dentary Crest | Dorsoventral height | 87.1 |
| Dentary Crest | DCH/MRH ratio *sensu* Vullo *et al.*, 2012 | 5.31 |
| Dentary Crest | Angle between posterior margin of dentary crest to rami | 91˚ |
| Hyoid | Max preserved length | 119.3 |

## Description and comparisons

**Generalities.** The skull (Table 2; Figs 2–5) is exposed in left lateral profile, revealing most elements of its antorbital portion and parts of the temporal and occipital portions. It bears a notably well-preserved soft-tissue crest that considerably extends dorsally, but does not extend caudally beyond the occiput. The palatal (Fig 2B) and most the occipital (Fig 3) regions of the skull are covered by sediment and could only be assessed through CT data. The skull articulates with an edentulous lower jaw, which bears a well-pronounced, anteriorly positioned dentary crest. At the anterior part of the premaxillomaxillae and dentaries, remnants of a keratinous rhamphotheca form narrow patches of soft tissue that extend over the bony limits of the rostrum.

Virtually the entire vertebral series is present (Table 3; Figs 6–8), including the atlas/axis complex and some caudal elements. Most vertebrae are in anatomical position, preserved in

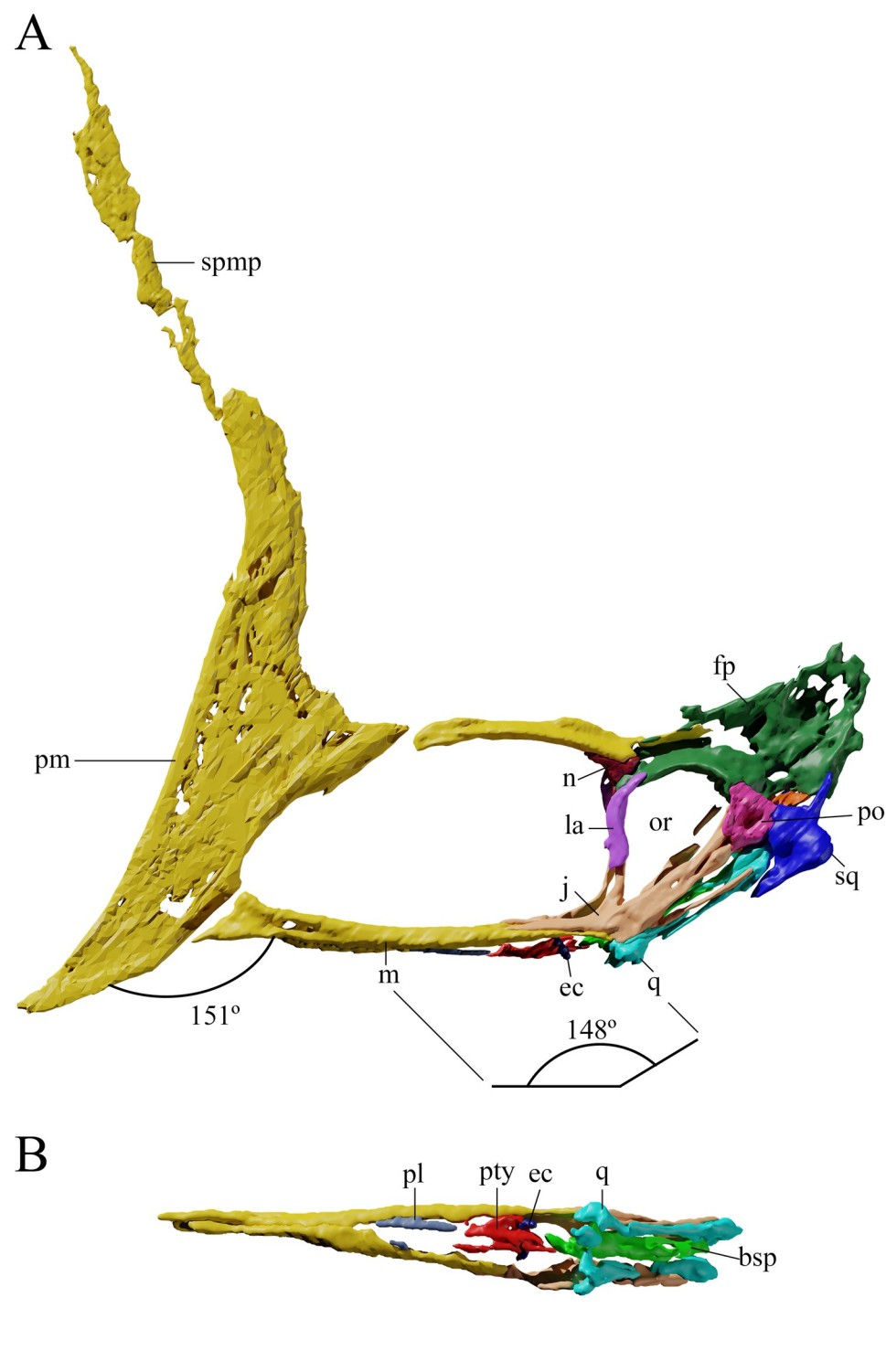

**Fig 2. *Tupandactylus navigans* GP/2E 9266 skull.** 3D model in left lateral view (A); palatal view (B). Abbreviations: ec, ectopterygoid; fp, frontoparietal; j, jugal; la, lacrimal; m, maxilla; n, nasal; or, orbit; pl, palatine; pm, premaxilla; po, postorbital; pty, pterygoid; q, quadrate; spmp, supra-premaxilllary bony process; sq, squamosal. Scale bars = 50 mm.

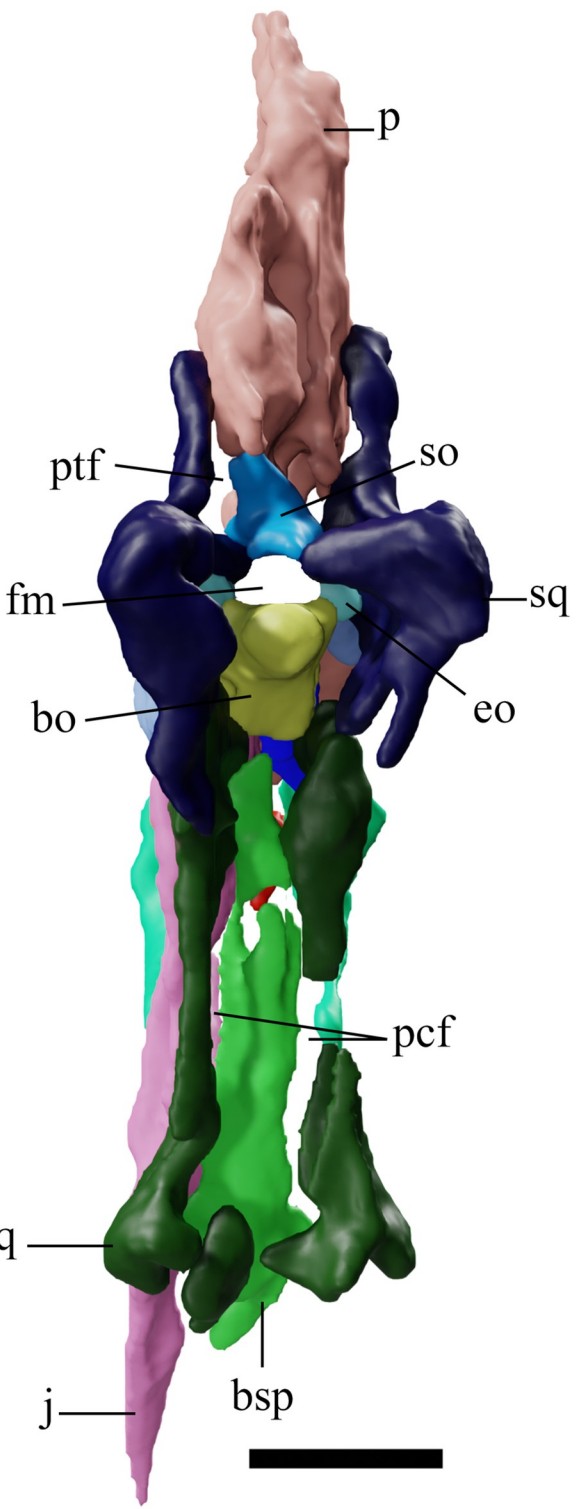

**Fig 3. *Tupandactylus navigans* GP/2E 9266 skull.** 3D model of occipital region of the skull. Abbreviations: bo, basioccipital; bsp, basisphenoid; fm, foramen magnum; eo, exoccipital; j, jugal; p, parietal crest; pcf, postcranial fenestra; ptf, posttemporal fenestra; q, quadrate; so, supraoccipital; sq, squamosal. Scale bar = 50 mm.

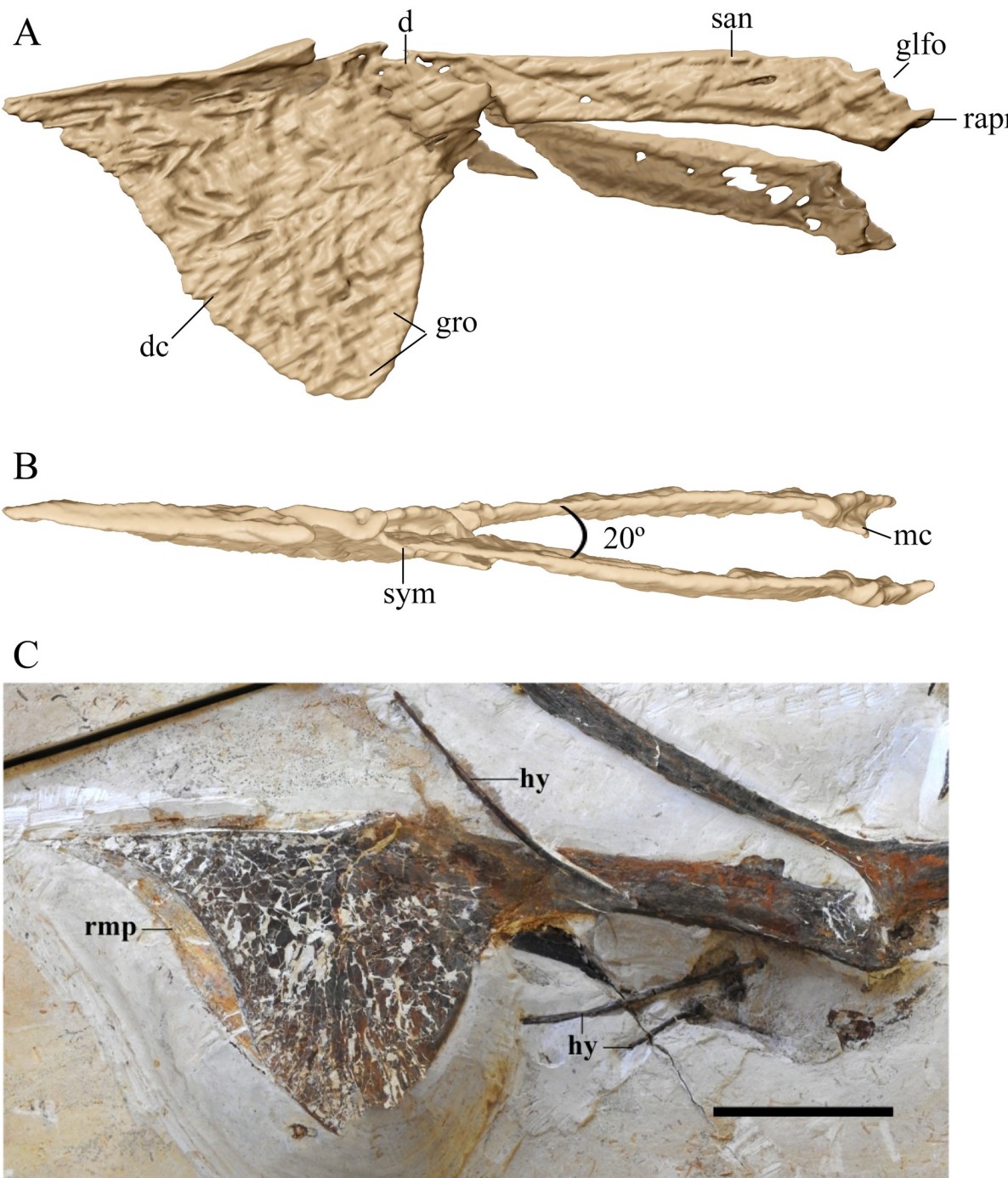

**Fig 4. *Tupandactylus navigans* GP/2E 9266 mandible.** 3D model in left lateral view (A); dorsal view (B); photograph in left lateral view (C). Abbreviations: d, dentary; dc, dentary crest; glfo, glenoid fossa; hy, hyoids; mc, medial cotyle; rapr, retroarticular process; rmp, rhamphotheca; san, surangular; sym, symphysis. Scale bars = 50 mm.

left lateral view. The caudal series, however, is slightly displaced and it is not clear whether its elements are rotated along their longitudinal axes. The cervical series is well preserved, but the centra of some elements are laterally crushed. On the other hand, most dorsal vertebrae are strongly weathered, so that their external bone layers are indistinguishable, exposing chunks of

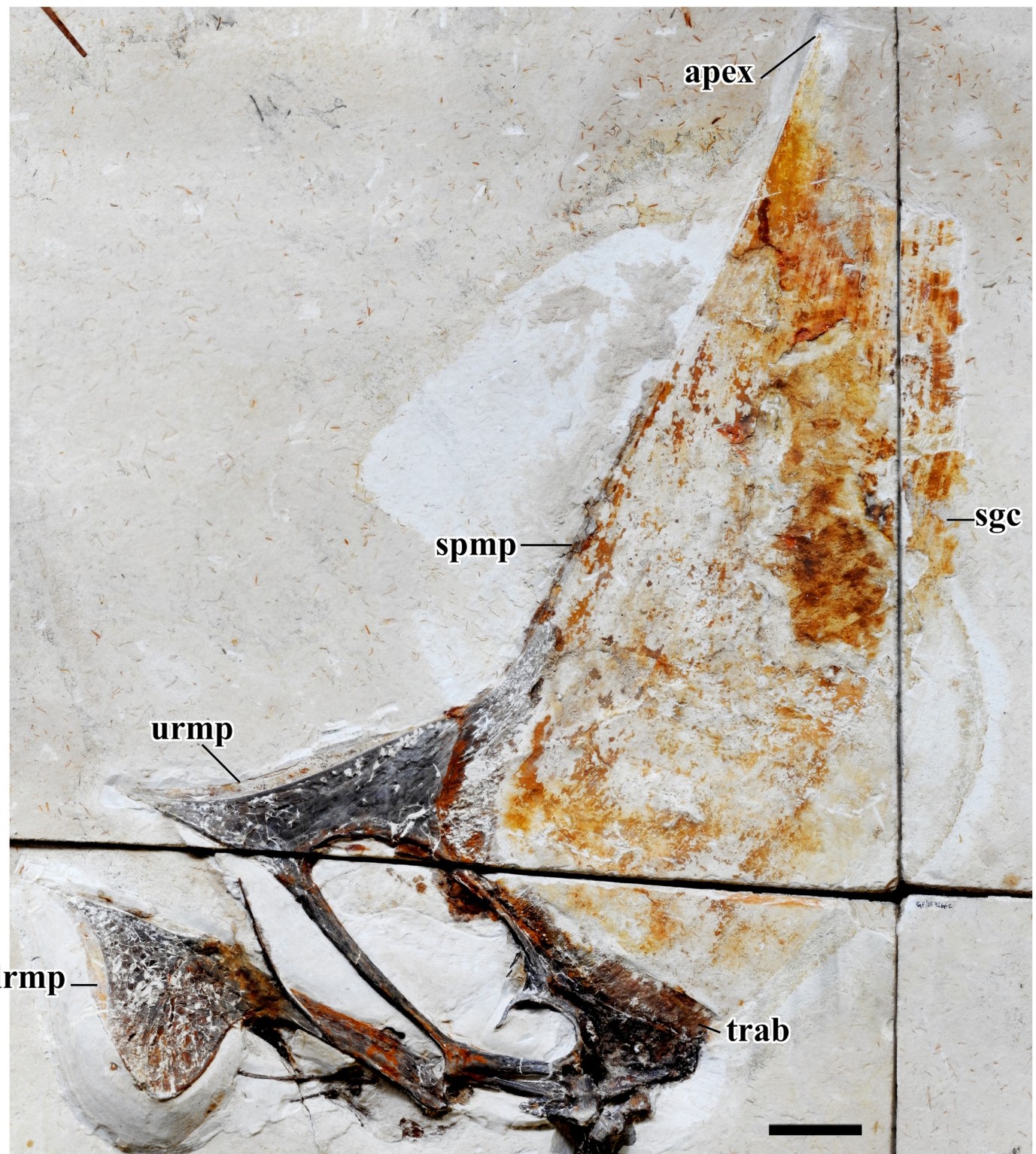

**Fig 5. *Tupandactylus navigans* GP/2E 9266 non-ossified tissue.** Photograph soft-tissue elements. Abbreviations: apex, apical-most point; lrmp, lower jaw rhamphotheca; sgc, sagittal crest; spmp, supra-premaxillary bony process; trab, trabecular bone; urmp, upper jaw rhamphotheca. Scale bar = 50 mm.

**Table 3. *Tupandactylus navigans* GP/2E 9266 axial skeleton measurements.**

| Bone | Comments | Measurement (mm) |
|---|---|---|
| Atlantoaxis | Centrum length | 21.7 |
| Atlantoaxis | Dorsoventral height | 40.3 |
| Atlantoaxis | Centrum width | 12.7 |
| Cervical Vertebra 3 | Centrum length | 38.2 |
| Cervical Vertebra 3 | Dorsoventral height | 21.1 |
| Cervical Vertebra 3 | Centrum width | 27.0 |
| Cervical Vertebra 4 | Centrum length | 49.9 |
| Cervical Vertebra 4 | Dorsoventral height | 21.5 |
| Cervical Vertebra 4 | Centrum width | 16.6 |
| Cervical Vertebra 5 | Centrum length | 46.7 |
| Cervical Vertebra 5 | Dorsoventral height | 21.7 |
| Cervical Vertebra 5 | Centrum width | 15.3 |
| Cervical Vertebra 6 | Centrum length | 52.0 |
| Cervical Vertebra 6 | Dorsoventral height | 24.7 |
| Cervical Vertebra 6 | Centrum width | 10.3 |
| Cervical Vertebra 7 | Centrum length | 45.4 |
| Cervical Vertebra 7 | Dorsoventral height | 27.6 |
| Cervical Vertebra 8 | Centrum length | 40.2 |
| Cervical Vertebra 8 | Dorsoventral height | 24.8 |
| Cervical Vertebra 9 | Centrum length | 23.7 |
| Cervical Vertebrae | Cervical vertebrae total length | 317.8 |
| Dorsal Vertebrae | Free dorsal vertebrae centrum average length | 14.2 |
| Dorsal Vertebrae | Free dorsal vertebrae centrum average width | 15.2 |
| Dorsal Vertebrae | Dorsal and sacral vertebrae total length | 211.9 |
| Caudal Vertebrae | Caudal vertebrae total length | 34.1 |
| Sternum | Anteroposterior length | 132.2 |

trabecular bone. Nine free vertebrae are present anterior to the notarium, with the first and second vertebrae fused into the atlas/axis complex. The five mid-cervical vertebrae (3–7) are comparatively long anteroposteriorly, with the typical condition displayed by cervical elements of azhdarchoids. Together with the atlas/axis complex, which is partially covered by the skull, these are here regarded as typical cervicals. Cervical vertebrae 8 and 9 share several features with dorsal elements and are here considered as cervicalized dorsal vertebrae. The large size of individual cervical vertebrae makes the cervical series comparable in size with the sum of the lengths of the dorsal and sacral series. As preserved, some mid-cervical and anterior dorsal vertebrae are partially covered by forelimb elements. For practical reasons, all pre-notarial free elements are considered as cervical vertebrae for the description below. The five anteriormost dorsal vertebrae constitute the notarium. Besides post-notarial free dorsal vertebrae, five posterior elements of the dorsal series are fused into the synsacrum. A large sternum (Fig 9) is also preserved as a plate-like bone just below the vertebral column.

The fore and hind limbs (Table 4; Figs 10–13) are not articulated with the girdles, but their elements remain in articulation with one another, preserving even complete autopodia. The wingspan is estimated to reach 2.7 m (measured by the sum of the total length of the humerus, ulna, metacarpal 4, and wing phalanges). The estimated wingspan and general measurements of the appendicular skeleton in tapejarids is found bellow (Table 5). Based on the fusion of the skeleton (i.e., sacrum, the proximal extensor processes of the first wing

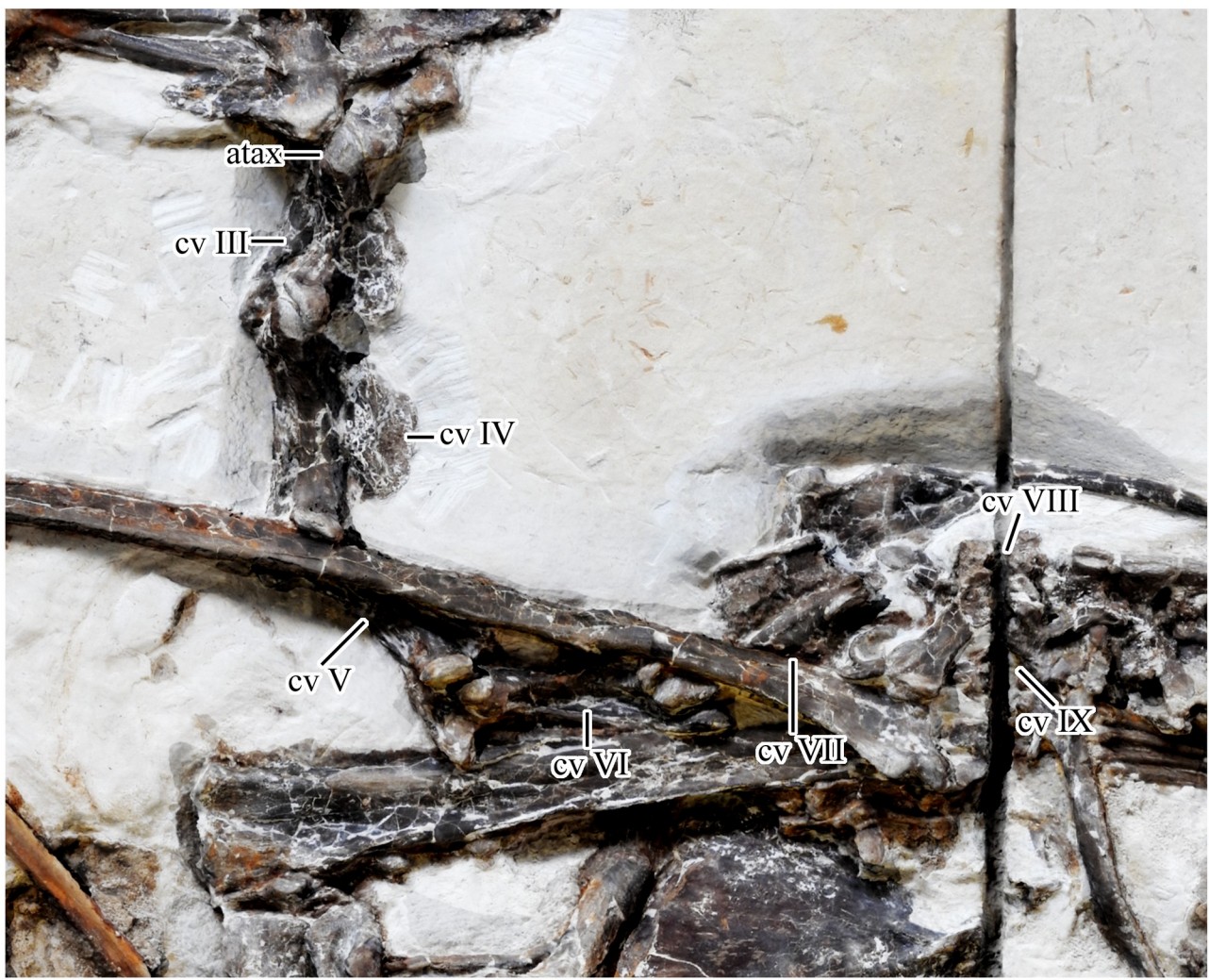

**Fig 6. *Tupandactylus navigans* GP/2E 9266 cervical vertebrae.** Abbreviations: atax, atlas-axis complex; cv, cervical vertebrae. Scale bars = 10 mm.

phalanges and the fusion of the notarium), GP/2E 9266 is here interpreted as an adult individual (see Discussion).

## Skull

**Premaxillomaxilla.** It is not possible to distinguish the premaxilla from the maxilla, as there is no visible suture between these elements. This is a common feature in pterosaurs and was previously observed in other tapejarids [10,22,23]. The premaxillomaxillae are slightly concave anteriorly, forming a sharp, ventrally oriented rostrum, and extending dorsally to form the premaxillary crest. A sharp rostral tip is commonly observed in tapejarine tapejarids (e.g., *Ta. wellnhoferi*, *Tupandactylus* sp., *Caiuajara dobruskii*, *Sinopterus dongi* [24], *Europejara olcadesorum* [2], and *Eopteranodon lii* [25]). The Rostral Value of GP/2E 9266 (as measured from the tip of the premaxillomaxillae to the anterior margin of the nasoantorbital fenestrae [26,27]) is 2.57, an intermediate value when compared with Chinese tapejarids (ranging from

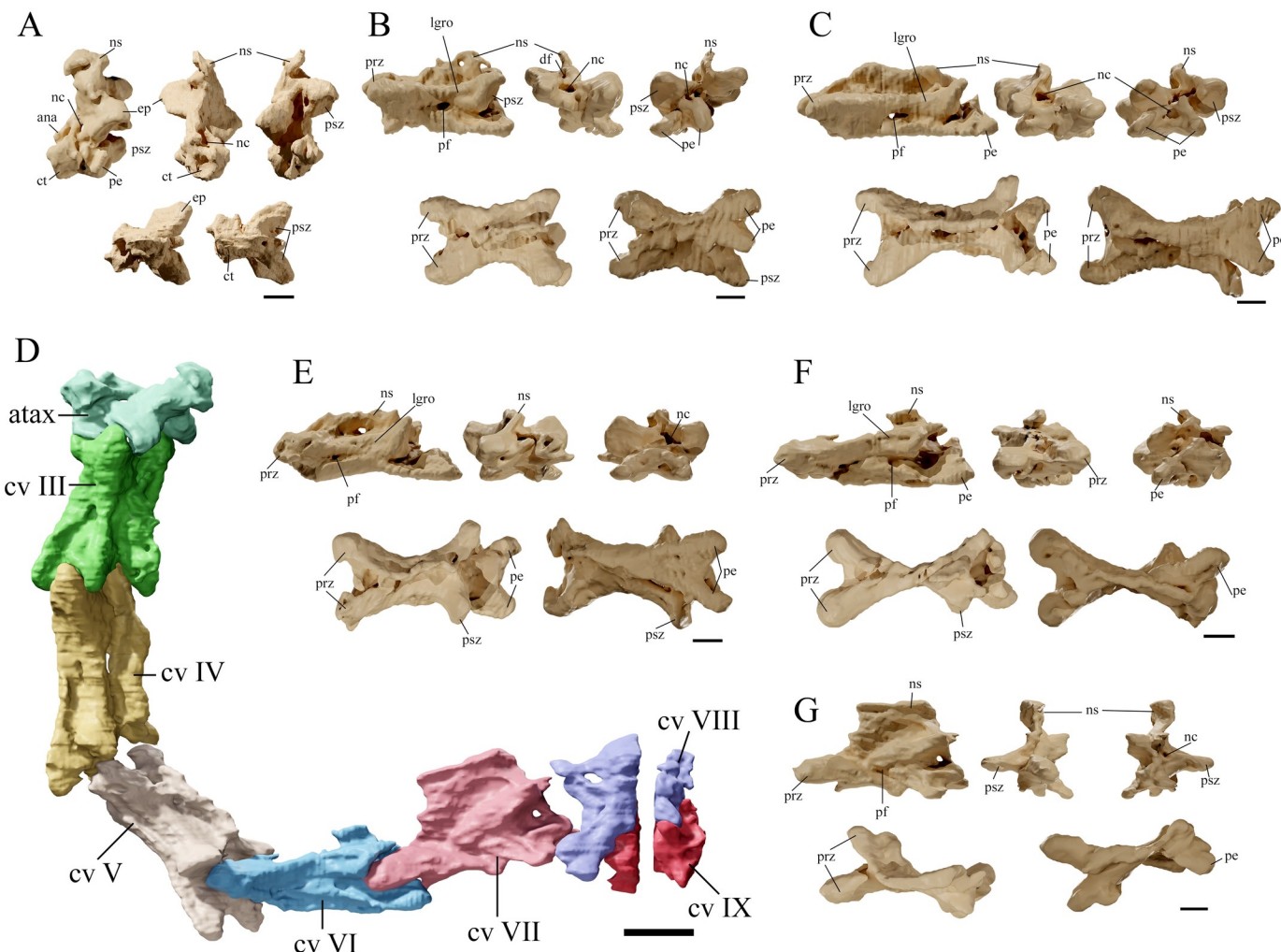

**Fig 7. *Tupandactylus navigans* GP/2E 9266 cervical vertebrae.** 3D model of atlas/axis complex (A); cervical vertebra 03 (B); cervical vertebra 04 (C); cervical vertebrae series (D); cervical vertebra 05 (E); cervical vertebra 06 (F); cervical vertebra 07 (G). From upper left to bottom right: left lateral view, anterior view, posterior view, dorsal view and ventral view. Abbreviations: ana, atlantal neural arch; atax, atlas-axis complex; ct, cotyle; cv, cervical vertebrae; ep, epipophysis; fo, pneumatic foramina; nc, neural canal; ns, neural spine; prz, prezygapophysis; pe, postexapophysis; psz, postzygapophysis. Scale bars = 10 mm.

1.76 in "*Huaxiapterus jii*" [23] to 4.05 in *S. lingyuanensis* [7,28]). The rostral index (RI *sensu* [29]) of GP/2E 9266 is 0.46, differing from those of the holotype (SMNK PAL 2344) and a referred specimen (SMNK PAL 2343) of *Tupa. navigans* (0.65 and 0.6, respectively), due to the premaxillary crest expansion. Low RI values are reported for short-rostra pterosaurs, contrary to the condition found in long-jawed forms, such as azhdarchids (RI values, 4.36–7.33, *sensu* [29]). GP/2E 9266 shares with other tapejarines a downturned anterior end of the rostrum, with a slope of 151° (Fig 2A). A short and ventrally deflected rostrum is synapomorphic of tapejarine tapejarids [1].

The premaxillomaxillae are posteriorly concave, delimitating the roughly semi-circular anterior margin of the nasoantorbital fenestrae. The maxillae extend posteriorly as thin horizontal plates, reaching the jugals below the orbits and forming the ventral margins of the nasoantorbital fenestrae, as well as a considerable portion of the palate. Dorsally, the palate is concave, with maxillary posterolateral processes confining the palatal plate in a deep concavity. For a discussion on the participation of the maxillae in the pterosaur palate, see [30,31].

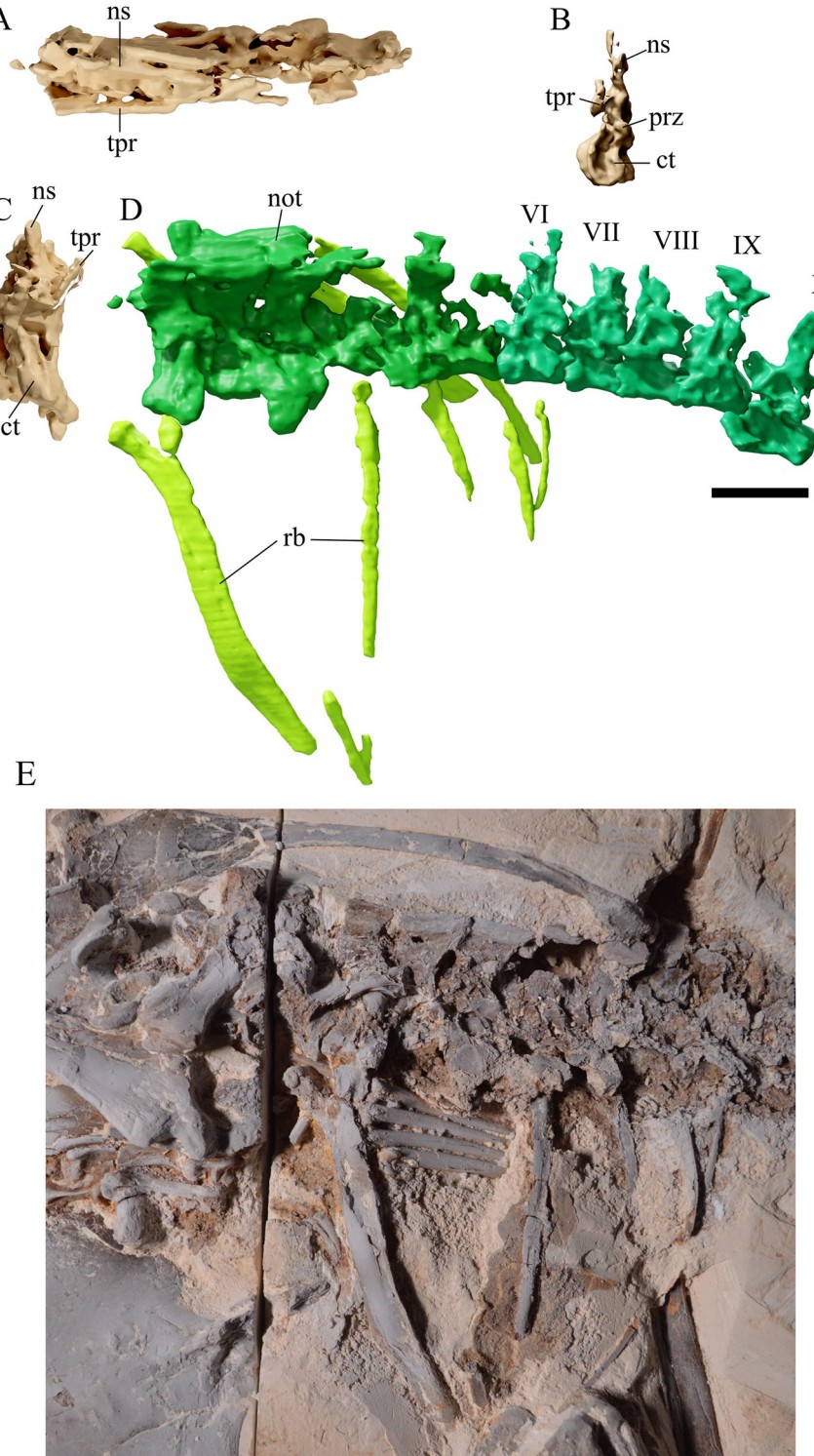

**Fig 8. *Tupandactylus navigans* GP/2E 9266 dorsal vertebrae and ribs.** 3D model of notarium in dorsal view (A); dorsal vertebra 06 in anterior view (B); notarium in anterior view (C); dorsal vertebrae series in left lateral view (D); photograph of notarium (E). Abbreviations: ct, cotyle; not, notarium; ns, neural spine; prz; prezygapophysis; rb, ribs; tpr, transverse process. Scale bar = 20 mm.

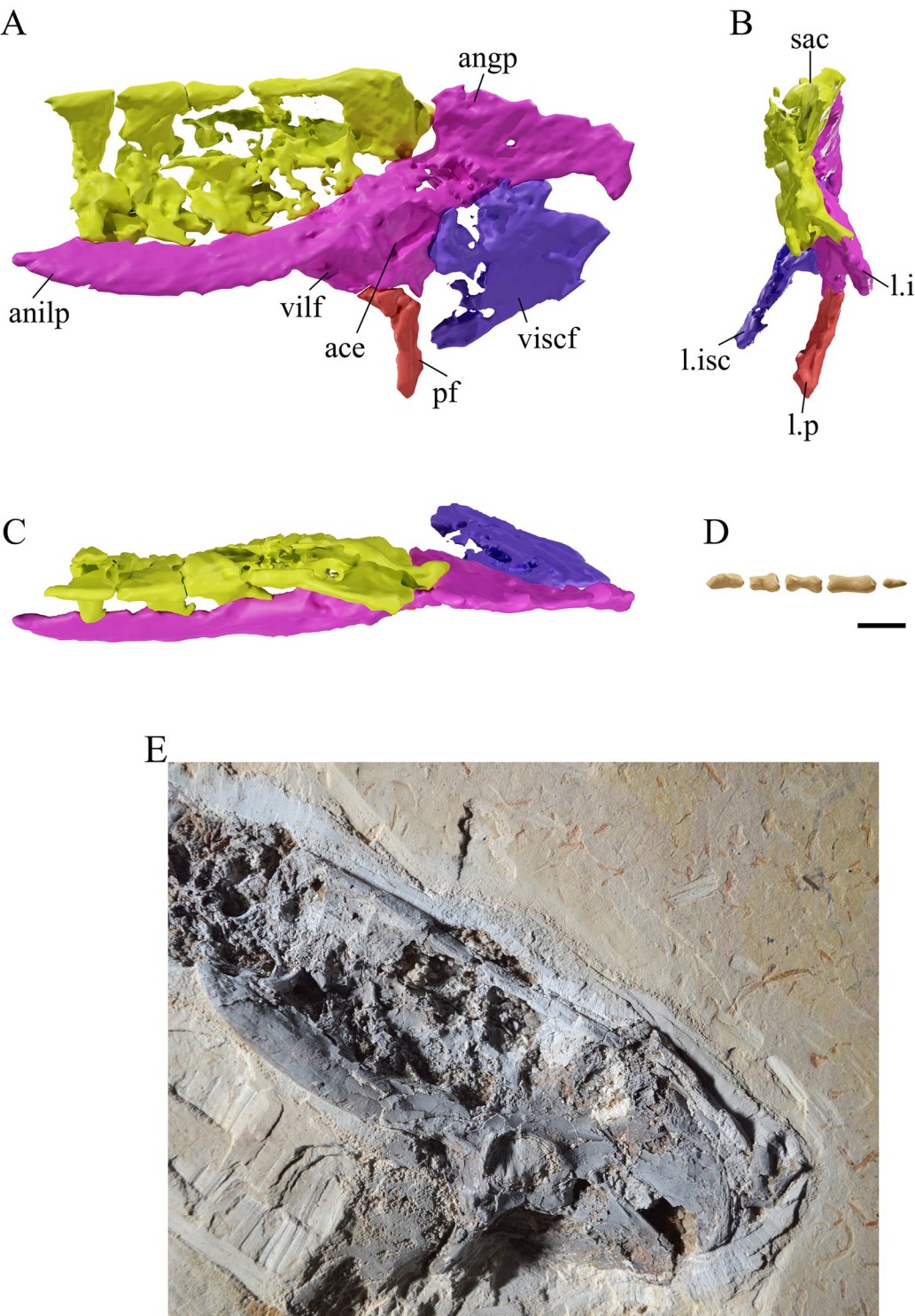

**Fig 9. *Tupandactylus navigans* GP/2E 9266 sacrum and caudal vertebrae.** 3D model of sacrum in left lateral view (A); anterior view (B); dorsal view (C); posterior caudal vertebrae in left lateral view (D); photograph of sacrum (E). Abbreviations: ace, acetabulum; angp, angular process; anilp, anterior ilium process; il, ilium; isc, ischium; p, pubis; sac, sacral vertebrae; vilf, ventral ilium fossa; viscf, ventral ischium fossa. Scale bar = 10 mm.

**Table 4.** *Tupandactylus navigans* **GP/2E 9266 appendicular skeleton measurements.**

| Bone | Comments | Measurement (mm) |
|---|---|---|
| Scapula | Length | 130.5 (l); 132.4 (r) |
| Scapula | Mid-shaft diameter | 8.0 (l); 10.8 (r) |
| Coracoid | Length | 83.5 (l); 86.8 (r) |
| Coracoid | Mid-shaft diameter | 8.5 (l); 5.8 (r) |
| Humerus | Length | 125.5 (l); 131.0 (r) |
| Humerus | Mid-shaft diameter | 15.5 (l); 19.5 (r) |
| Humerus | Deltopectoral length | 22.5 (l); 25.8 (r) |
| Humerus | Deltopectoral width | 39.9 (l); 30.6 (r) |
| Ulna | Length | 186.1 (l); 188.1 (r) |
| Ulna | Mid-shaft diameter | 13.7 (l); 16.6 (r) |
| Radius | Length | 180.6 (l); 184.1 (r) |
| Radius | Mid-shaft diameter | 10.1 (l); 9.42 (r) |
| Pteroid | Length | 97.0 (r) |
| Metacarpals 1–3 | Max length | 82.1 (l); 60.8 (r) |
| Metacarpal 4 | Length | 183.9 (l); 181.7 (r) |
| Metacarpal 4 | Mid-shaft diameter | 18.5 (l); 16.5 (r) |
| Manual Phalanx 1 (r) | Length | 25.6 (d1); 15.7 (d2); 27.6 (d3) |
| Manual Phalanx 2 (r) | Length | 24.6 (d2); 4.6 (d3) |
| Manual Phalanx 3 (r) | Length | 24.4 (d3) |
| Manual Unguals (r) | Length | 27.3 (d1); 24.5 (d2); 26.4 (d3) |
| Manual Unguals (r) | Max depth | 11.2 (d1); 10.1 (d2); 9.3 (d3) |
| Wing phalanx 1 | Length | 317.5 (l); 304.0 (r) |
| Wing phalanx 1 | Mid-shaft diameter | 10.5 (l); 12.2 (r) |
| Wing phalanx 2 | Length | 193.1 (l); 196.1 (r) |
| Wing phalanx 2 | Mid-shaft diameter | 7.1 (l); 6.5 (r) |
| Wing phalanx 3 | Length | 123.1 (l); 129.9 (r) |
| Wing phalanx 3 | Mid-shaft diameter | 4.1 (l); 4.2 (r) |
| Wing phalanx 4 | Length | 39.3 (l) |
| Wing phalanx 4 | Mid-shaft diameter | 2.6 (l) |
| Ilium | Preacetabular length | 78.3 (l) |
| Ilium | Postacetabular process length | 44.3 (l) |
| Pubis | Depth | 26.2 (l) |
| Femur | Length | 167.9 (l); 163.1 (r) |
| Femur | Mid-shaft diameter | 14.2 (l); 13.6 (r) |
| Femur | Angle between head and shaft | 46° (r) |
| Tibia | Length | 246.2 (l); 248.8 (r) |
| Tibia | Mid-shaft diameter | 11.7 (l); 10.6 (r) |
| Fibula | Length | 74.2 (l); 67.2 (r) |
| Metatarsals (l) | Length | 35.8 (d1); 44.9 (d2); 47.2 (d3); 46.3 (d4) |
| Metatarsals (l) | Width | 2.6 (d1); 2.4 (d2); 2.5 (d3); 3.0 (d4) |
| Pes Phalanx 1 (l) | Length | 19.2 (d1); 6.4 (d2); 8.5 (d3); 9.4 (d4) |
| Pes Phalanx 2 (l) | Length | 16.5 (d2); 3.5 (d3); 5.4 (d4) |
| Pes Phalanx 3 (l) | Length | 14.1 (d3); 11.9 (d4) |
| Pes Unguals (l) | Length | 11.6 (d1); 14.1 (d2); 12.2 (d3); 10.7 (d4) |
| Pes Unguals (l) | Max depth | 4.2 (d1); 4.7 (d2); 5.5 (d3); 5.1 (d4) |

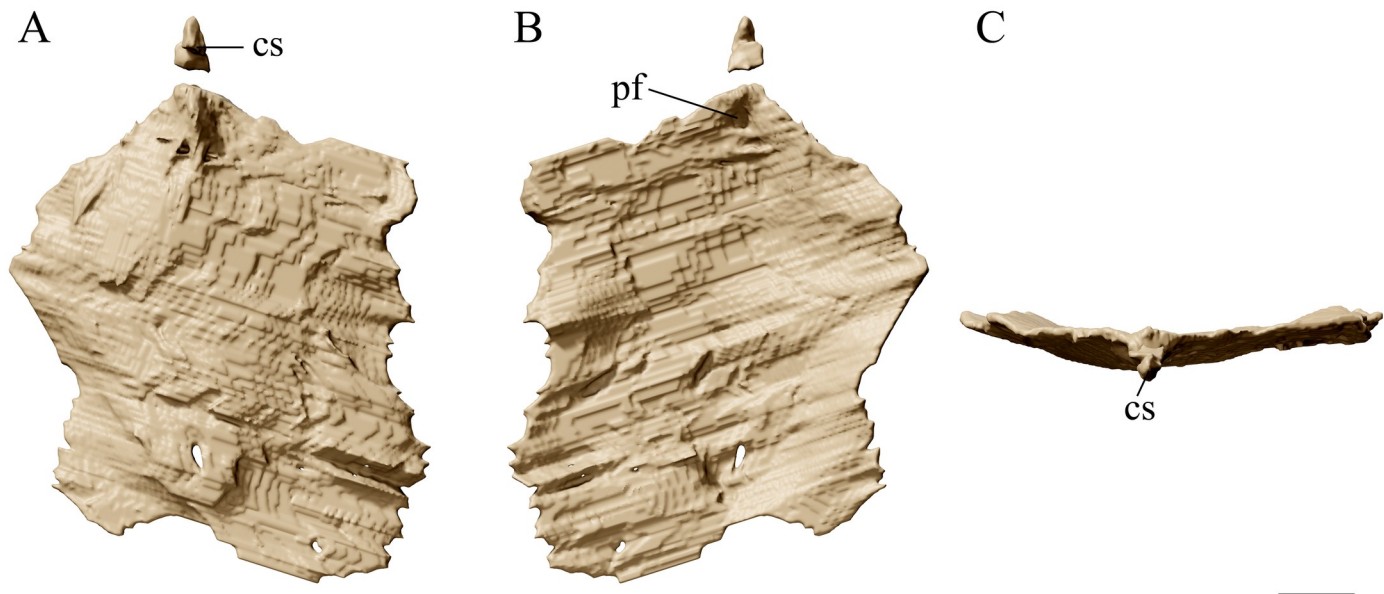

**Fig 10. *Tupandactylus navigans* GP/2E 9266 sternum.** 3D model of sternum in ventral view (A); dorsal view (B); anterior view (C). Abbreviations: cs, cristospine; pf, pneumatic foramina. Scale bar = 20 mm.

The presence of a prominent premaxillary crest is shared by all tapejarids. In GP/2E 9266, the exposed bone component of the crest is triangular in lateral view, with a sharp dorsal tip, as was previously observed in *Tupa. imperator* and other *Tupa. navigans* specimens, differing from the rounded crest of *Ta. wellnhoferi* and the expanded blade of *C. dobruskii*. However, CT data revealed that an anteriorly deflected expansion is present at the most dorsal part of the premaxillary main body (Fig 2A), resembling the premaxillary crest of early ontogenetic stages

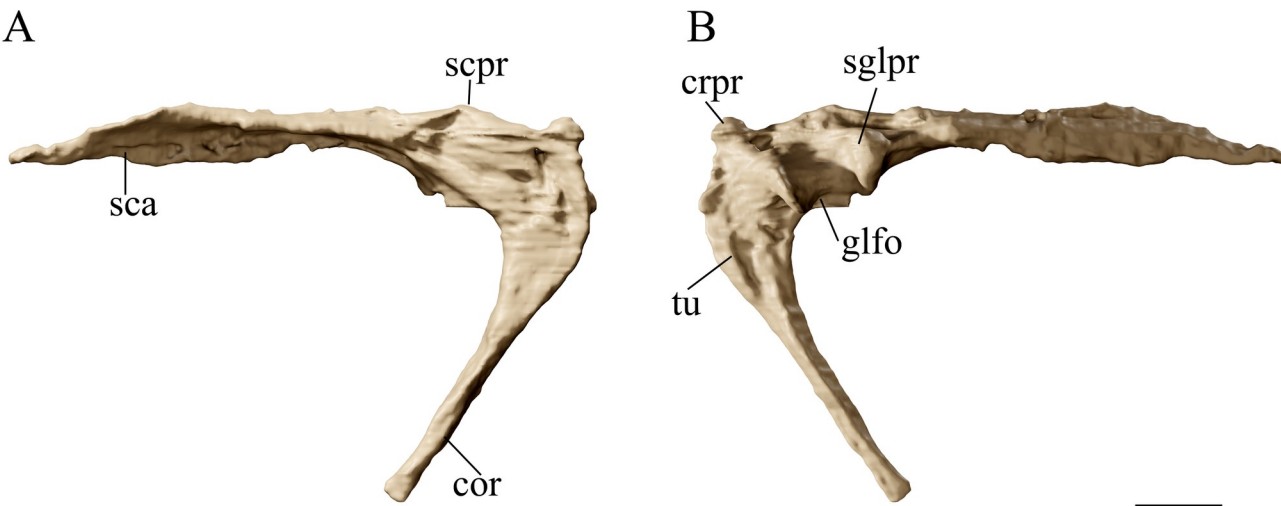

**Fig 11. *Tupandactylus navigans* GP/2E 9266 left scapulocoracoid.** 3D model of left scapulocoracoid in posterior view (A); anterior view (B). Abbreviations: cor, coracoid; crpr; coracoid process; glfo, glenoid fossa; sca, scapula; scpr, scapula process; sglpr, supraglenoid process; tu, coracoid tuberculum. Scale bar = 20 mm.

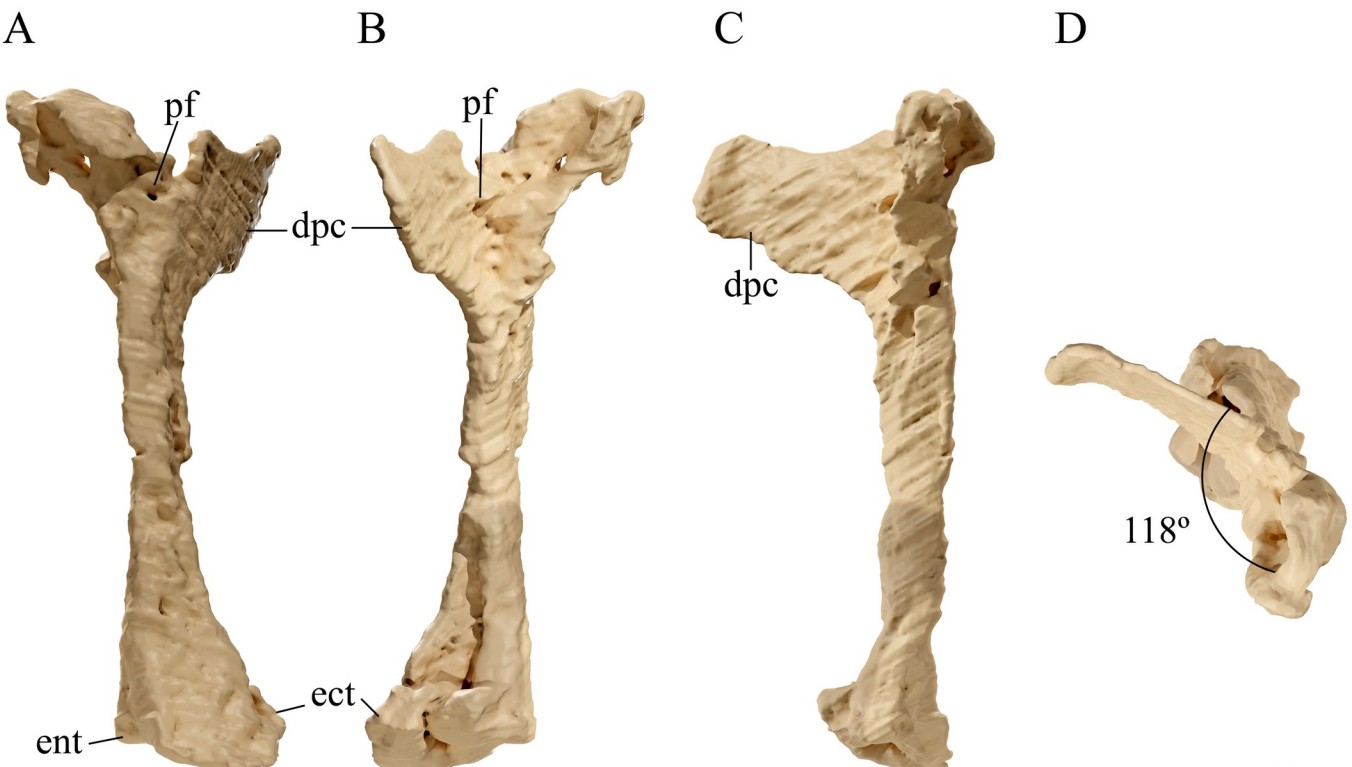

**Fig 12. *Tupandactylus navigans* GP/2E 9266 right humerus.** 3D model of right humerus in dorsal view (A); ventral view (B); posterior view (C); medial view (D). All views are in flight position. Abbreviations: dpc, deltopectoral crest; ect, ectepicondyle; ent, entepicondyle; pf, pneumatic foramen. Scale bar = 10 mm.

of *C. dobruskii* [5]. This expansion is covered by sediment and is possibly also present in the holotype of *Tupa. navigans* (SMNK PAL 2344). As the remaining premaxillomaxilla, the premaxillary crest is perforated by foramina and has thin grooves probably associated with blood vessels.

The slender and tall supra-premaxillary bony process (Fig 2A) starts at the dorsal tip of the main bone component of the premaxillary crest and extends dorsally, forming the anterior margin of the soft tissue crest. This structure is dorsally broken in GP/2E 9266, with its top represented only by an impression. A similar bony process is present in *Tupa. imperator*. In this form, however, the process deflects posteriorly, contrasting with the perpendicular condition in *Tupa. navigans* [9,10].

### Nasals

The nasals (Fig 2A) are triangular-shaped bones that delimit the dorsoposterior margin of the nasoantorbital fenestra. Lateroventrally they contact the lacrimals, and posteriorly the frontals. Their dorsal limits articulate with the posterodorsal processes of the premaxillae, which form the anterodorsal and dorsal margins of the nasoantorbital fenestrae. The suture between nasals and premaxillae cannot be established with confidence. In most tapejarids, this suture seems to be reduced or absent on the lateral surface of the skull [32] (but see specimen CPCA 3590 [10]).

### Lacrimals

The lacrimals (Fig 2A) are very thin bones that limit the nasoantorbital fenestrae posteriorly and the orbits anteriorly. They are anteriorly perforated by large foramina, encompassed

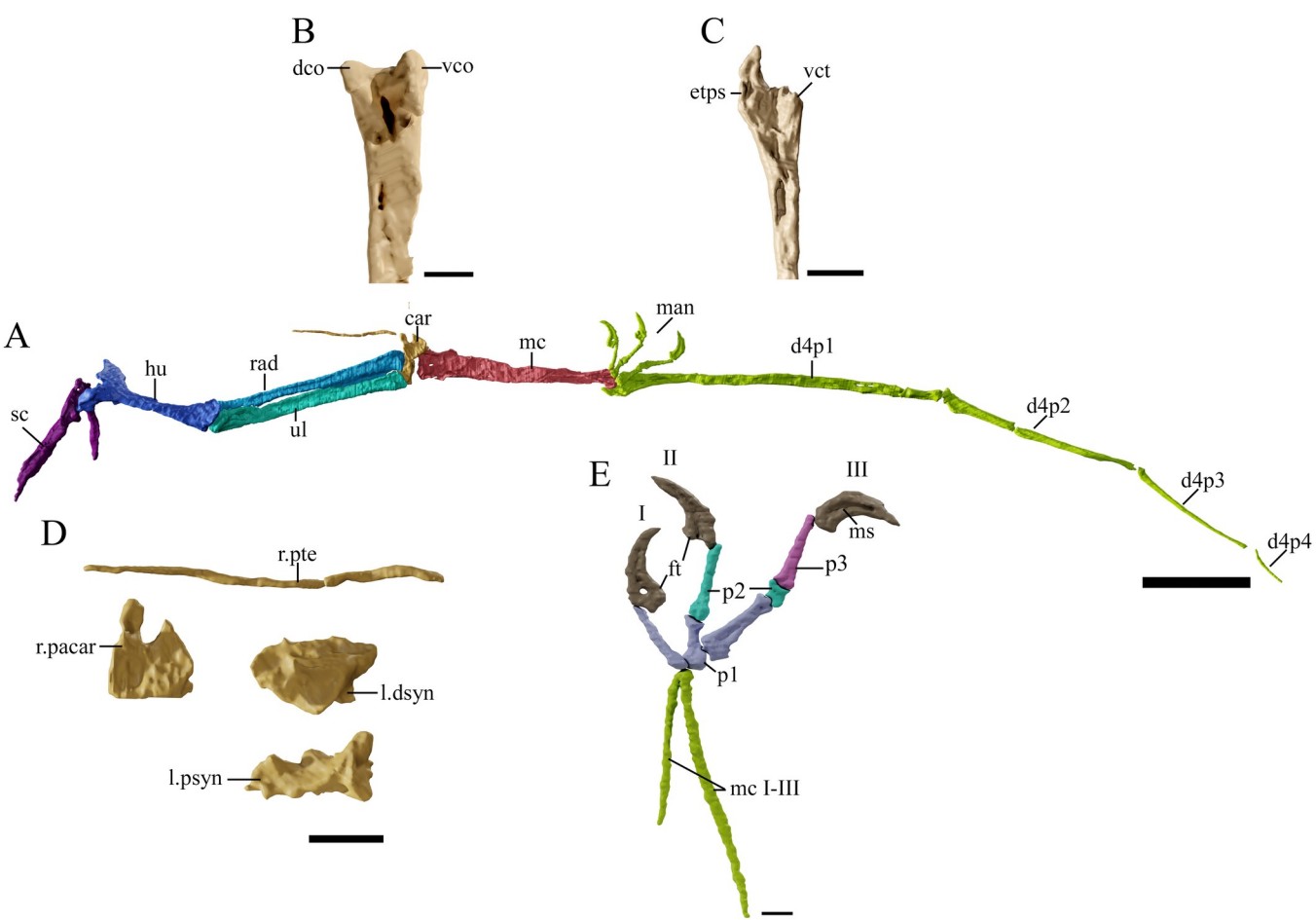

**Fig 13. *Tupandactylus navigans* GP/2E 9266 forelimb elements.** 3D model of forelimb in dorsal view (A); distal left metacarpus IV in posterior view (B); right proximal wing phalanx I in ventral view (C); left carpal elements in anterior view (D); left manus in lateral view (E). Abbreviations: car, carpus; d1-4, wing phalanges; dco, dorsal condyle; dsyn, distal syncarpal; etps, extensor tendon process saddle; ft, flexor tubercle; hu, humerus; man, manus; mc1-4, metacarpals 1–4; ms, medial sulcus; p1-3, manual phalanges; pacar, paraxial carpal; psyn, proximal syncarpal; pte, pteroid; rad, radius; sc, scapulocoracoid; ul, ulna; vco, ventral condyle; vct, ventral cotyle. Scale bar = 50 mm (A); 10 mm (B-E).

between their lateral processes and their main rami. In this respect, GP/2E 9266 resembles the condition displayed by *C. ybaka* and *S. dongi*, but is very different from the highly perforated lacrimals of *Ta. wellnhoferi*.

## Frontoparietals

Frontals and parietals (Fig 2A) are fused and laterally compressed in GP/2E 9266. The external surface of the left frontoparietal is partially eroded, but its general outline is preserved. The frontals contact the postorbitals laterally and delimit the dorsal margins of the orbits and the anterior margins of the supratemporal fenestrae. The orbits are positioned high in the skull, with their dorsal portions almost at the same level as the nasoantorbital fenestrae. This is also the condition observed in *Ta. wellnhoferi*, *C. dobruskii* and *Tupa. imperator*, but differs from the low orbits of *C. ybaka* and thalassodromine tapejarids. Medially, the parietals compose the inner and posterior margins of the supratemporal fenestrae. The frontoparietals deflect poster-odorsally to form short and blunt crests that do not extend posteriorly further than the

**Table 5. Estimated wingspan for tapejarids and general measurements of the appendicular skeleton for tapejarids.** All measurements are in (mm) unless stated otherwise.

| Taxon | *Tupa. navigans* GP/2E 9266 | *Ta. wellnhoferi* SMNK PAL 1157 | *C. dobruski* CP.V 872a | *S. dongi* IVPP V 13363 | *'H' corollatus* ZMNH M8131 |
|---|---|---|---|---|---|
| Estimated wingspan (ew) | 2.7 m | 1.23 to 1.3 m | 2.35 m* | 1.2 m | 1.5 m |
| Humerus length (hu) | 131 | 72 | 84 | 59 | 80 |
| Radius length (ra) | 184 | 91 | - | 84 | 110 |
| Ulna length (ul) | 188 | 96 | 110 | 88 | 114 |
| Metacarpal 4 length (mc4) | 184 | 101 | - | 95 | 152 |
| First wing phalanx length (fwp) | 318 | 158 | - | 121 | 176 |
| Second wing phalanx length (swp) | 196 | 63 | 121 | 88 | 109 |
| Femur length (fe) | 168 | 68 | 84 | 74 | 93 |
| Tibia length (tb) | 249 | 96 | 103 | 104 | 155 |
| hu/fe ratio | 0.78 | 1.05 | 1.00 | 0.80 | 0.86 |
| ra/hu ratio | 1.40 | 1.27 | - | 1.42 | 1.38 |
| ul/hu ratio | 1.44 | 1.34 | 1.31 | 1.49 | 1.43 |
| mc4/hu ratio | 1.40 | 1.41 | - | 1.61 | 1.91 |
| fwp/hu ratio | 2.43 | 2.21 | - | 2.05 | 2.21 |
| fwp/swp ratio | 1.62 | 2.50 | - | 1.38 | 1.62 |
| tb/fe ratio | 1.48 | 1.40 | 1.23 | 1.41 | 1.68 |

* indicates maximum wingspan estimation in [5].

posterior limits of the squamosals. This differs from the posteriorly elongated frontoparietal crests of *Ta. wellnhoferi* and, especially, *Tupa. imperator*.

## Jugals

As in other tapejarids (i.e., *Ta. wellnhoferi*, *C. dobruskii*) and in the holotype of *Tupa. navigans* (SMNK PAL 2344), the jugal is a tetraradiate bone (Fig 2A). Its posterior processes participate in both the upper and lower margins of the lower temporal fenestra. The maxillary process delimits the posteroventral margin of the nasoantorbital fenestra. The lacrimal process is thin and slightly deflected posteriorly, forming an angle of little more than 90˚ with respect to the maxillary ramus.

## Quadrates

The quadrates are posteriorly inclined in an angle of 148˚ (Fig 2A), falling within the 125˚-150˚ range displayed by other tapejarids (e.g., *Thalassodromeus sethi* ~125˚; *Tupuxuara leonardii* ~130˚; *Ta. wellnhoferi* ~140˚; *Tupa. imperator* ~145˚; *C. ybaka* ~150˚) [32]. This bone is slender medially, but forms a deep convex articular surface with the mandible.

## Squamosals

The squamosals (Fig 2A) are broad elements of the temporal region of the skull, forming the dorsal limits of the infratemporal fenestrae and, together with the postorbitals, the lateral surfaces of the supratemporal fenestrae. Anteriorly, they contact the postorbitals over a round and shallow foramen. The main bodies of the squamosal are broad, with medial flanges that turn into dorsally and ventrally expanded processes. The dorsal process of each squamosal is short and slender when compared to the elongate anteroventrally oriented process. This latter

extends parallel to the quadrate, as in *Ta. wellnhoferi* [33] and "*Tupu.*" *deliradamus* (SMNK PAL 6410, [34]).

## Supraoccipital

The supraoccipital (Fig 3) forms a considerable part of the occipital area, expanding from the dorsal margin of the foramen magnum to the posteriormost ventral tip of the sagittal crest. Its contact surfaces with squamosals, parietals, opisthotics, and exoccipitals are not clear. As in other pterodactyloids, the supraoccipital plate is broad and well developed [33], medially limiting the posttemporal fenestrae.

## Basioccipital

The occipital condyle (Fig 3) is broad, with a rounded ventral portion and a straight dorsal surface. The basioccipital fuses ventrally with the basisphenoid, forming a plate that encompasses most of the ventral area of the occiput. This is also the condition observed in *Ta. wellnhoferi* [33], forming a well-developed basisphenoid plate also observed in *C. ybaka* [32].

## Palate

The palatal region is covered by sediment and could only be assessed through CT-Scanning (Fig 2B). Posteriorly, palatines, pterygoids, and basipterygoids are preserved, all of them displaying a considerable degree of lateral compression. The maxillae are ventrally fused, forming a palatal plate that extends from the rostrum to the level of the anterior limits of the nasoantorbital fenestrae (see [31]). No foramina could be observed on the ventral surface of the maxillary plate, though this is probably a consequence of insufficient resolution of the CT data. Similar to what was previously observed in *Ta. wellnhoferi*, the palate of GP/2E 9266 is concave anteriorly (matching the downslope of the premaxillomaxillae), becoming convex posteriorly. The posterior convexity follows a discrete lateral expansion of the premaxillomaxillae close to the anterior margins of the nasoantorbital fenestrae and makes the GP/2E 9266 posterior palate visible in lateral view. In addition, GP/2E 9266 lacks the deep palatal ridge characteristic of *Tupu. leonardii* [31,35,36]. The anterior margins of the choanae are slightly convex, and the vomers could not be identified.

The palatines apparently form the lateral margins of the choanae and the pterygo-ectopterygoid fenestrae. These bones are slender and anteroposteriorly long. Their contacts with the pterygoids are not discernible, what is common among azhdarchids [31].

The ectopterygoids are well developed in GP/2E 9266 and contact the maxillary bars anteriorly. As in *Pteranodon longiceps* [37] and azhdarchoids (see [31]), the ectopterygoids cross the pterygoids dorsally, with an angle of ~25˚ with respect to the anterior rami of the pterygoids. The contact between ectopterygoids and pterygoids occurs in the medial process of the latter.

The pterygoids are dorsoventrally thin. The lateral processes of these bones divide the suborbital fenestrae in two, with elongated and subtriangular pterygo-ectopterygoid fenestrae anteriorly and oval, shorter suborbital fenestrae posteriorly. Both fenestrae can be observed in lateral view of the skull. The same condition was previously reported for *Tupu. leonardii* [31].

## Mandible

The edentulous lower jaw is complete (Fig 4) and preserved still in articulation with the left quadrate. The left hemimandible is exposed, while the right hemimandible could be accessed only through CT-Scanning. The latter is ventrally deflected, probably reflecting a pre-burial breakage. The symphysis is located on the first half of the mandible, accounting for 41% of its

length. Tapejarine tapejarids present this element accounting for less than 50% of total mandibular length, with *Ta. wellnhoferi* bearing a lower value for symphyseal length relative to the total length of mandible (38%) [38].

The symphysis is slightly downturned anteriorly. Close to the deepest region of the dentary crest, the dorsal surface of the symphysis projects dorsally, leveling with the dorsal margins of the hemimandibles. The hemimandibles are laterally compressed. The right element is broken and deflected downwards, with minor degree of lateral distortion. This allows the measurement of the angle between both hemimandibles (~20˚). This angle falls in the range of *Ta. wellnhoferi* (24˚, based on AMNH 24440) and the azhdarchoid *Jidapterus edentus* (20˚ for the holotype CAD-01; [39]). It is, however, below the ~30˚ of *Aymberedactylus cearensis* [38]. The preserved rhamphotheca covers the anterior dorsal concavity of the symphysis, extending ventrally to encase the anterior border of the dentary crest.

Although suture lines between mandibular bones are hard to determine laterally, the dentary was probably the largest element, with the characteristic step-like dorsal margin observed in tapejarines and bearing a deep bony crest. This later extends from the anterior edge of the mandible to almost its midportion, ending in a straight posterior margin that forms an angle of about 90˚ with the long axis of the lower jaw. The mandibular crest morphology is variable within the Tapejaridae. The ratio between dentary crest height and hemimandible height (see [2] for DCH/MRH) is 5.3 in GP/2E 9266, the highest among tapejarines (2.2 in *S. dongi*; 2.5 in *Ta. wellnhoferi*; 3.0 in *Tupa. imperator*; 4.0 in *E. olcadesorum*). In *Ta. wellnhoferi* this crest is shallower and slightly displaced posteriorly, but still does not surpass the midportion of the jaw, thus shorter relative to the condition of GP/2E 9266 and *Tupa. imperator* [10]. *S. dongi*, "*Huaxiapterus*" *corollatus* [40], and "*Huaxiapterus*" *benxiensis* [41] bear low, blade-like, but longer dentary crests. The dentary crest bears shallow grooves that radiate from its centre to its distal margins. This was first observed for *Tupa. imperator* (CPCA 3590) and may represent deep vascularization of this area or simply be related to the keratinous rhamphotheca anchorage.

The hemimandibles are elongated and shallow, with a height/length ratio of 0.071, a little over *Ay. cearensis* and below half the ratio of *Ta. wellnhoferi* (0.142). As in *Tupa. imperator*, the retroarticular process is short and blunt, differing from the elongated process of *Ay. cearensis*.

## Hyoid apparatus

The first pair of ossified ceratobranchials is elongated and slender (Fig 4C). This element is slightly bowed medially, with an elevated posterior half as in *E. olcadesorum* [2].

## Cranial non-ossified tissue

**Rhamphotheca.** Sheaths of tissue forming a rhamphotheca (Figs 4C and 5A) cover the anterior and ventral borders of the premaxillomaxillae, as well as the anterior portion of the dentary (extending to the ventral dentary crest). These sheaths have already been figured for *Tupa. navigans* holotype (SMNK PAL 2344), as well as for a referred specimen (SMNK PAL 2343) of this taxon [9] and were interpreted as a keratinous rhamphotheca. This structure, the outline of which closely matches that of the rostrum of these specimens, has also been observed in *Tupa. imperator* [10,42], and inferred for *C. dobruskii* due to presence of abundant lateral and palatal foramina on the external surface of the premaxillomaxillae [5]. Similar foramina are visible in GP/2E 9266 covering the anterior region of the premaxillomaxillae.

## Sagittal crest

The eye-catching prominent soft-tissue crest (Fig 5B) exceeds five times the skull height at its dorsal most preserved point. The crest is formed by the dorsally and anteriorly striated bones of the premaxillomaxillae, together with a large portion of soft tissue sustained anteriorly by the elongated, vertically projected supra-premaxillary process. A notable point is that a well-marked range on the inclination of the anterior margin of this crest is reported for young (~115˚) to adult (up to ~90˚) individuals in *C. dobruskii*, which means that this crest becomes steeper as the animal grows [4]. This trend cannot be confirmed for *Tupandactylus*, due to the lack of an ontogenetic series for the genus. However, if we assume this tendency, it would be suggestive of an advanced maturity for individuals with steeper dorsal crests. The fact that all known *Tupa. imperator* specimens have caudally oriented crests and larger skulls than known *Tupa. navigans* (GP/2E 9266, SMNK PAL 2344 and SMNK PAL 2343) may confirm a posteriorly deflected crest as diagnostic for *Tupa. imperator*.

The soft-tissue cranial crest can be divided in two regions: a ventral fibrous portion and a dorsal smooth part. The sub-parallel vertical pattern of *Tupa. navigans* fibers [9] differs from that observed in *Tupa. imperator* [10] for its slight anterior orientation (posteriorly oriented in the latter), with no signs of cross-over. The fibrous crest borders the dorsal region of the skull, extending from the base of the supra-premaxillary crest to the posterior end of frontoparietals. It projects upwards until the transition of these striae into the soft-tissue median crest. This differs from what is reported for *Tupa. imperator*, the fibers of which contact directly the smooth region of the crest [1,10].

The smooth crest is posteriorly convex, with some patches of darker perpendicular tissue preserved. Those patches are more apparent at the dorsal portion of the crest and bear a striate pattern. It is worth mentioning that, in GP/2E 9266, this portion of the crest ends dorsally in a concave notch.

## Axial skeleton

**Cervical vertebrae.** Only the posterior part of the atlas/axis complex (Figs 6 and 7A) is exposed, other parts are covered by the squamosal and occipital bones and were only assessed through CT-Scanning. The axial neural spine has a tapering dorsal margin, similar to what is displayed on *Ta. wellnhoferi* (SMNK PAL 1137) and *Tupu. leonardii* (IMCF 1052). Although the preservation of the atlas-axis complex is rare among the Pterodactyloidea, in taxa like *Pteranodon* sp. (YPM 2440), *Anhanguera piscator* (NSM-PV 19892), *Anhanguera* sp. (AMNH 22555), and *Azhdarcho lancicollis* (ZIN PH 105/44) the axis has a dorsally tapering neural spine, terminating in a round surface [43–46]. The atlas and the axis of GP/2E 9266 are fused (Fig 7A), with an intervertebral foramen beneath the atlantal neural arch. Apart from the neural spine, the axial neural arch displays a dorsoventrally deep left postzygapophysis, which is dorsally continuous to a round tubercle. The axial centrum presents well developed postexapophyses, as is visible ventral to the left prezygapophysis of cervical vertebra 3.

Cervical vertebrae 3–6 (Figs 6 and 7B–7G) have proportionally long centra (length to width ratio ranging from 3.0 to 5.0), albeit not reaching the extreme condition exhibited by azhdarchids (i.e., length to width ratio = 12 for *Quetzalcoatlus* sp. and ranging from 5.3 to 8.1 in *Az. lancicollis* [46]). The anteroposterior length of the centra increases towards the posterior midcervical series, reaching its maximum in cervical vertebrae 6. Cervical vertebrae 7–9 (Figs 6, 7D and 7G) have shorter centra, but this observation is hindered by the fact that cervical vertebrae 8 and 9 are poorly preserved. A single small pneumatic foramen is present on the lateral surface of the midcervical centra. Lateral pneumatic foramina commonly pierce the centra of thalassodromine cervical vertebrae [47–49]. As an example, large pneumatic openings of

cervical vertebrae 2–3 were recognized for a tapejarid specimen (AMNH 22568) by [48]. In contrast, these structures seem to occur more rarely in tapejarines, with a single small foramen being documented for a midcervical of *Ta. wellnhoferi* [6] and two for another tapejarid specimen (MN 4728-V) [48]. Lateral cervical foramina are also rare among the Azhdarchidae, but were reported for cervical vertebra 8 of *Az. lancicollis* [46]. On the other hand, very large pneumatic foramina in the lateral surface of the centra are widespread among pteranodontoids (*sensu* [50,51]).

The cervical centra are strongly procoelous, as the cotyles extend posteriorly beyond the postzygapophyses, and are easily discernible in lateral aspect. The cotylar region of the centrum of all elements (including the atlas-axis complex) is laterally expanded, with well-developed postexapophyses. In these elements, the ventral margin of the centrum is slightly concave. As previously observed, the centra of tapejarine tapejarids cervical vertebrae display concave ventral edges, whereas thalassodromine tapejarids have cervical centra with straight ventral margins [48].

As is typical among the Pterodactyloidea, cervical neural arches are laterally swollen. Although the precise position of the neurocentral sutures is unclear, a step-like longitudinal ridge roughly separating the centrum end neural arch is visible in lateral aspect. The cervical neural arches are spool-shaped in dorsal view, as the pre- and postzygapophyses consistently diverge laterally from a comparatively constricted midportion. This is a common feature among dsungaripteroids (*sensu* [50]), displayed, for instance, by *Pteranodon* sp. (YPM 2730; [45]), *Anhanguera* sp. (AMNH 22555; [43]), *Ta. wellnhoferi* (SMNK PAL 1137; [6]), *'Phobetor' complicidens* (GIN125/1010 [52,53]), and *Tupu. leonardii* (IMCF 1052). Both pre- and postzygapophyses are prominent and bear wide articular facets. The lateral surfaces of the postzygapophyses display longitudinal grooves starting near the midpoint of the neural arches and terminating in a region adjacent to the articular surfaces. These longitudinal grooves are particularly deep in cervical vertebrae 3 and 5 and, to our knowledge, are exclusive for GP/2E 9266 across dsungaripterids. Epiphyses (sometimes referred as postzygapophyseal tubercles, e.g. [54]) are conspicuously present in cervical vertebrae 3 to 7. Among the postaxial elements of the cervical series, neural spines are completely exposed in cervical vertebrae 3, 4, and 7, with all (except the latter) displaying trapezoid neural spines with modestly convex dorsal margins. Moreover, in both cervical vertebrae 3 and 4, the neural spine has an anterodorsally sloping anterior margin, whereas the posterior one is subvertical. Similar hatched-shaped neural spines were previously reported for indeterminate thalassodromines and tapejarines (e.g., AMNH 24445 and AMNH 22568 [48]) and is also present in *Tupu. leonardii* (IMCF 1052). Neural spine morphology differs sharply in cervical vertebra 7, in which it assumes an anteroposteriorly long and dorsoventrally deep rectangular shape. In addition, the neural spine of cervical that vertebra thickens dorsally to form a spine table similar to those displayed by anterior (notarial) dorsal vertebrae. Cervical vertebrae 7–9 display anteroposteriorly short centra compared to those of midcervical elements, which indicates a transitional morphology between typical cervical and dorsal vertebrae or, as proposed by [40] for *Pteranodon* sp., that these are cervicalized dorsal vertebrae.

## Dorsal vertebrae

The dorsal series can be divided into three distinct regions: i) five anterior vertebrae forming a notarium (Fig 8A and 8C–8E), ii) five free mid dorsal vertebrae, and iii) five synsacral dorsal vertebrae. Robust transverse processes are visible in dorsal vertebrae 1, 4, and 5, all three still in association with their corresponding ribs. The neural spines of the three anteriormost dorsal vertebrae are exceptionally well developed, with subvertical anterior and concave posterior

margins. They are in very close association to one another and display transversely thick, spine table-like dorsal ends with the postzygapophyses fused to the prezygapophyses of the subsequent vertebra. This is probably due to ossified tendons associated to the notarial ossification. The centra of the first five dorsal vertebrae are fused, although the two posteriormost of these lack expanded and fused neural spines. This condition was also observed in *Tupu. leonardii* (IMCF 1052), where only the first three neural spines are fused [55]. This pattern of fusion (i.e., fusion of the centrum followed by fusion of the neural spines) was interpreted as an intermediate stage of notarium development (NS3) [55]. The notarium in GP/2E 9266 lacks the supraneural plate, as is common for azhdarchids [55]. It is worth mentioning that *C. dobruskii* apparently does not bear a notarium [5], but this structure was reported in an indeterminate tapejarid from the Crato Formation (MN 6588-V) [19].

Very few relevant features are available regarding the mid free dorsal vertebrae (Fig 8B and 8D). They are anteroposteriorly shorter than cervical/anterior dorsal vertebrae, and the better-preserved ones display tall neural spines that differ from those of the first five dorsal vertebrae by the absence of a thickened spine table and for having concave anterior and posterior margins. Thoracic vertebrae previously described for *Ta. wellnhoferi* (SMNK PAL 1137) display either neural spines with subvertical anterior and posterior margins, or convex and concave anterior and posterior margins ([6], Fig 5).

Synsacral vertebral (Fig 9A–9C and 9D) fusion seems to be similar to that observed in notarial elements, in which the co-ossification of neural spines results from bundles of ossified tendons restricted to the dorsal limits of these structures. Poor preservation prevents an accurate assessment of possible fusion between pre/postzygapophyses and centra of successive synsacral vertebrae, but it is clear that the synsacrum was formed by five individual elements. The Crato Formation tapejarid (MN 6588-V) [19] has a similar synsacral configuration as GP/2E 9266.

## Caudal vertebrae

Five spindle-shaped caudal vertebrae (Fig 9D) lie in close association with the posterior elements of the pelvic girdle. They are mainly featureless, not presenting neural arches. The two presumably anterior elements are robust, slightly longer than wide. The posterior caudal vertebrae are three times longer than wide (ranging from 5–6.1 mm long vs. 1.8–2.1 mm wide). It is unlikely that these five elements represent the whole caudal series, but the poor preservation of azhdarchoid tails hinders a proper evaluation of vertebral count for representatives of this clade.

## Sternum

The sternum (Fig 10) is displaced from its anatomical position and exposed in ventral view. Some appendicular elements overlap both lateral margins of this bone, which is also partially covered by rock matrix. The sternal plate is wide and roughly square-shaped. A large pneumatic foramen pierces the dorsal surface of the sternum, posterior to the cristospine. This feature was also observed in *Ta. wellnhoferi* (SMNK PAL 1137; [6]). Despite being dorsoventrally compressed, the sternum appears to have been considerably convex in its ventral surface. The concave anterolateral margins of the sternal plate converge to contact the cristospine, which is comparatively as long anteroposteriorly as that of *Tupu. leonardii* (IMCF 1052), and longer than that of *Ta. wellnhoferi* (SMNK PAL 1137). The posterior margin of the sternal plate appears to be straight, similar to that of an indeterminate tapejarid (MN 6558-V). *Ta. wellnhoferi* (SMNK PAL 1137) and *Tupu. leonardii* (IMCF 1052) present sternal plates with distinctly convex posterior margins.

## Appendicular skeleton and girdles

**Scapulocoracoids.** Both scapulocoracoids are present in GP/2E 9266, but the left one is better preserved (Fig 11). It lies in dorsal aspect close to the distal portion of the left hindlimb. The scapulocoracoids are single functional elements formed by the fusion of scapulae and coracoids without any sign of suture lines. A transverse fracture is visible on the proximal part of the coracoidal shaft in the left scapulocoracoid. Yet, this does not represent the suture line with the scapula, as this articulation is positioned more dorsally so that both bones contribute to the glenoid fossa (see [43], Fig 16). The scapula is considerably longer than the coracoid, as is common among azhdarchoids (e.g. [19,47]). It expands lateromedially close to the articulation with the coracoid, so that a round supraglenoid process is visible dorsal to the procoracoid in anterior view. Distal to this, the scapular shaft twist along its axis, which makes the scapular blade to face laterally. This blade expands progressively towards its distal end, which is broken on the left element, but preserved on the right one, showing a flat articular surface for the notarium. The procoracoid is visible as a well-developed lateral bulge in both scapulocoracoids. The proximal portion of the coracoid expands ventrolaterally to form a deep coracoidal flange as in *Pt. longiceps* (e.g., YPM 2525; [45]). The indeterminate Crato Formation tapejarid (MN 6588-V) also presents a ventrolateral coracoidal flange, posterior to which a modest tubercle is visible, as in GP/2E 9266. Thalassodromine tapejarids present much deeper coracoidal tubercles when compared with MN 6588-V, what can also be observed in other specimens (e.g., AMNH 22567 and MN 6566-V) [47,49]. The glenoid fossa is deep and lies between the scapular and coracoidal tubercles.

## Humerus

The left humerus is exposed in dorsal view and is dorsoventrally flattened, whereas the right element is preserved in anterior view, with its proximal portion still covered by rock matrix and by the left forelimb phalanges. The left humerus is divided in two slabs and is nearly complete, lacking some of its distal portion, whereas the right humerus (Fig 12) is complete. GP/2E 9266 humeri are relatively small when compared to those of other azhdarchoids (humerus length/femur length = 0.78, see Table 5) [56,57].

Both deltopectoral crests remain covered by sediment and were also assessed through CT data. The crest is well developed and forms ~90° of the humeral shaft. The left deltopectoral crest is anteroposteriorly compressed and the right one is complete, unflatten, and twice as long as it is tall. This structure is slightly curved anteroventrally, as observed in *Ta. wellnhoferi* [6]. The right humeral head is broad and posteroventrally oriented, contrarily to the dorsal oriented caput of *Ta. wellnhoferi* [6]. A large subcircular pneumatic foramen is visible beneath the humeral head, on the dorsal margin of the bone, and another foramen is located ventrally, beneath the anterodorsal margin. The presence of both dorsal and ventral pneumatic foramina was observed in ornithocheiroids and *Ta. wellnhoferi*, but not in *S. dongi* and, thus far, other azhdarchoids [6,40] or pteranodontids [58].

The humeral shaft is straight and constricted medially, broadening to twice its width at its distal end, with an almost flat radioulnar articular surface. Its medial surface is not as anteroposteriorly compressed as it is in *Ta. wellnhoferi*, but is similar to that of *C. dobruskii*, and relatively thinner than the condition in earlier pterodactyloids [5,6,47,59].

The entepicondyle expands distally and posteriorly, forming the wider dorsodistal margin of the humerus. This expansion is about as broad as the ectepicondyle and not as strongly projected as in pteranodontids [58]. The ectepicondyle is slightly expanded anteriorly as is seen in other azhdarchids [60].

Distally, the ventral surface of the right humerus is flattened, while the dorsal surface is rounded, what gives a D-shaped aspect to the bone cross section. This pattern is characteristic of azhdarchoids, differing from the subtriangular-shaped condition of ornithocheirids [50,58]. The radial condyle is wider and more spherical than the ulnar condyle. Both condyles are separated by a deep pneumatic foramen medially located at the distal surface of the humerus.

## Radius and ulna

The left radius and ulna are exposed in ventral view and separated in their middle in different slabs. The mid/proximal portion of the ulna lies under the left humerus, whereas its mid/distal portion is partially covered by the right femur and tibia. The right radius is preserved in anterior view, whereas the right ulna has its dorsal surface exposed. The ulna is slightly longer than the radius, and both are longer than the humerus by 48% and 44%, respectively. Both ulna/humerus and radius/humerus length ratio are similar to *S. dongi* (see Table 5) [24].

Both radius and ulna are straight and gracile. The radius has slightly over half the mid-shaft diameter of the ulna (0.60 of the ulnar shaft diameter). This characteristic is shared with other tapejarids (0.57 for *C. dobruskii* CP.V 869, 0.69 for *Tupu. leonardii* IMCF 1052), whereas that relation is respectively close to and over 0.7 for pteranodontids and archaeopterodactyloids [5,50]. Although compressed, the ulnae show a gentle, dorsally oriented distal curvature, as found in *Ta. wellnhoferi* [6,61].

The ulna has a ventrally expanded prominent articular tubercle at its distal end. In pteranodontids and ornithocheirids, this tubercle is medially placed, but ventrally projected in other azhdarchoids and archaeopterodactyloids [58].

## Carpus and metacarpus

Two syncarpal bones, together with the paraxial carpal and the pteroid, form the carpal complex. The left carpals (Fig 13A and 13D) are exposed in anterior view, whereas the right carpals lie in posterior view, mostly covered by the right metacarpal IV. The proximal carpal is concave distally, with its midportion proximo-distally constricted and its medial region prominently expanded both proximally, as an articular facet for ulna, and distally, articulating with the distal carpal. Posteriorly, the proximal carpal forms a deep ridge. The distal carpal is a massive, sub-triangular bone that is convex at its proximal part, with an anterior articular tubercle, and almost flat, with a sharp medial condyle, at its distal portion.

The right pteroid (Fig 13A and 13D) is very slender and long, reaching over 0.50 of the ulnar length. Proximally, it has a noticeable posteriorly facing convexity, which gives it a curved and slender rod-like shape, as in other pterodactyloids, such as azhdarchoids, ornithocheirids, and pteranodontids [58,62].

Left metacarpals I-III (Fig 13E) are exposed, whereas those from the right side could only be accessed through CT-Scanning. They do not connect to the carpus, barely reaching the proximal half of metacarpal IV. This condition has been previously observed in pteranodontids and in other azhdarchoids, such as *S. dongi*, "*H.*" *corollatus* and *Quetzalcoatlus* sp. [24,40,63]. Metacarpals I-III are very slender, rod-like bones, with an acute proximal end and a broad articular tubercle on the distal end. The left metacarpal IV (Fig 12B) is exposed in ventral view, whereas the right metacarpal IV (Fig 13A) is exposed in dorsal view and is a long and slender bone. It comprises 0.37 of the "inner wing length", defined by the humerus, radius/ulna and metacarpal IV, and is relatively shorter than in "*H.*" *corollatus* (0.44; see [40]). The right metacarpal IV is longer than the humerus (1.46 of humeral length, see Table 5) and has virtually the same length as the ulna. Both proportions closely follow the condition displayed by *Ta. wellnhoferi* and other tapejarids, but are relatively reduced when compared to that of

azhdarchids (2.30 of humeral length in *Quetzalcoatlus* sp.). Metacarpal IV is 0.57 of the first wing phalanx length. Ornithocheirids have metacarpal IV measuring ~0.40 of the first wing phalanx, whereas apparently all known tapejarids have a ratio of ~0.60 [64]. The right metacarpal IV is proximally broad, constricted to almost half its width distally. The anteroproximal end is sharp, with a concave cotyle forming its anterior articular facet with the distal carpus. The proximal posterior end is rounded and forms the articular condyle. The distal end of the metacarpal IV is convex, with a steep anterior articular surface.

## Manual digits

All digits of the left manus are present and complete (Fig 13A and 13E), following the phalangeal formula 2-3-4. The ungual phalanges preserve a keratinous soft-tissue outline. The abductor tubercle of the ungual phalanx of each digit is broad and round. The first phalanx of digit III has the unusual morphology described for *Ta. wellnhoferi* [6], in which the proximal margin is formed by two proximal condyles separated by a large sulcus. This phalanx is the longest and broadest of the first three digits. This condition differs from that of *Ta. wellnhoferi*, in which the longest phalanx is the first of digit I. Distally, the first phalanx of digits II and III are expanded posteriorly, with flat articular surfaces.

The manual ungual phalanges are broad, present a strong distal curvature, and have over twice the length of the pedal ungual phalanges. Most other pterodactyloids are similar in this respect, with the exception of *Ta. wellnhoferi*, the manual ungual phalanges of which are less than twice as long as those of the pes. The ungual phalanges articulate with the double distal tubercles of the pre-ungual phalanges, with a broad and round flexor tubercle. A deep medial sulcus crosses the ungual phalanges from their distal tip to the proximal end, as noted in other tapejarids (e.g., SMNK PAL 1137).

The left first wing phalanx is partially broken at its proximal end, whereas the right one (Fig 13A and 13C) is complete and was preserved with its dorsal side exposed. The right first wing phalanx is 2.5 the length of the humerus, as in other tapejarids [6] (see Table 5). Medially, the shaft bows slightly and is anteroposteriorly constricted, with a small round expansion at its distal end. A small pneumatic foramen lies beneath the dorsal cotyle, at the anterior-proximal end, as in several other pterodactyloids, such as *Ta. wellnhoferi* SMNK PAL 1137, the pteranodontid SMU 76476, and *Az. lancicollis* ZIN PH 36/44 [6,46,65]. The prominent extensor tendon process forms a posteriorly oriented hook and is totally fused to the proximal region of the bone.

The second (Fig 13A) and third wing phalanges (Fig 13A) bear proximal and distal articular expansions as those of the first wing phalanx. The second wing phalanx is 0.61 the length of the first wing phalanx, with virtually the same length as the ulna. In other tapejarids such as *Ta. wellnhoferi* and *C. dobruskii*, these ratios are higher (see Table 5), but much smaller in azhdarchids (e.g., 0.50 in *Quetzalcoatlus* sp.). When comparing the third wing phalanx with the first one, the lengths are more departed. The right third phalanx is ventrodorsally compressed to a third of its anteroposterior width and is 0.38 the length of the first wing phalanx (0.65 in *Ta. wellnhoferi*, 0.96 in *C. dobruskii* and 0.29 in *Quetzalcoatlus* sp.). The fourth wing phalanx (Fig 12A) is the shortest and mainly featureless.

## Pelvis

The pelvis (Fig 9A–9C and 9E) is exposed in left lateral view and is almost complete, lacking only the prepubis and some minor parts of the postacetabular process of the pubis. No suture lines between individual elements are discernible, which is sometimes used as a proxy for ontogenetic maturity [66–69]. The preacetabular process of the ilium is very long and

machete-shaped, exposing its wider surface towards de lateral view of the fossil skeleton. As this faces dorsally in most pterosaurs (e.g. [67,68]), it is likely that lateral compression caused this projection to rotate laterally along its main axis in GP/2E 9266. The margin directed dorsally (presumably the medial margin of the preacetabular process) is slightly concave, extending almost parallel to the main axis of the sacral vertebral series. In contrast, the margin directed ventrally (assumed here to be the lateral one) is moderately convex. If the preacetabular process of the ilium indeed rotated laterally, it strongly resembles that of *Ta. wellnhoferi* (SMNK PAL 1137; [6]). Both margins of this process converge anteriorly to form a pointed tip. Its anatomical ventral margin thickens into a crest that dorsally limits a concave anterior flange of the pubis where the suture line between both bones would probably be located.

The ilium expands posteriorly to form a fan-shaped postacetabular process with a constricted base and a markedly convex posterodorsal margin that terminates in a thick, anteriorly expanded ridge. Although a fan-shaped postacetabular process with a constricted shaft is also present in some archaeopterodactyloids (e.g., *Pterodactylus antiquus*), the condition displayed by GP/2E 9266 appears to be restricted to azhdarchoids and is present, for instance, in *Ta. wellnhoferi* (SMNK PAL 1137; [6]) and an indeterminate tapejarid (MN 6588-V; [19]). The acetabulum is partially filled with rock matrix and comprises a wide oval aperture anterodorsally delimited by a moderately thick ridge.

The pubis is a short, blunt and anteroventrally directed element that connects posteriorly with the ischium to form the ischiopubic plate. The ischiopubic plate formation occurs in late ontogeny [69,70], and it appears not to have reached its maximum development in GP/2E 9266, given that its ventral margin is concave and the pubis stands out as an individualized element. The obturator foramen is wide and oval, with its long axis extending anteroposteriorly. It opens ventrally to the acetabulum, in a similar position to that displayed by MN 6599-V [19].

## Femur

Both femora are present, with the left element exposed in posterior view, and the right one (Fig 14A–14C) in dorsal view. These elements are ~1.28 the length of humerus in GP/2E 9266, and although this ratio is seen in most azhdarchoids, the femora are almost the same size as the humerus in *C. dobruskii* (see Table 5) and is disparate in azhdarchids such as *Zhejiangopterus linhaiensis* (1.48) [5,40,71,72].

The femoral head (assessed by CT data) forms a 134˚ angle with the shaft, a similar condition to the majority of pterosaurs except pteranodontids and ornithocheirids [71,73,74]. The femoral neck is constricted medially, and the round caput is prominent, as in other azhdarchoids and pteranodontids [58]. The greater trochanter is subtriangular shaped and bears a proximal expansion, but lacks the anterior expansion found in other azhdarchoids such as *Quetzalcoatlus* sp. and *Ta. leonardii*. The lateral trochanter expands distally, forming a round convexity near the fourth trochanter. A small pneumatic foramen lies between the greater trochanter and the femoral neck.

The femoral shaft bows posteriorly in lateral view but is straight is anterior and posterior views. As noted for other azhdarchoids, the fibular condyle expands posterodorsally and comprises most of the distal expansion of the femur. The distal intercondylar fossa is as prominent as in *Ta. wellnhoferi* and thalassodromine tapejarids [6,47].

## Tibia and fibula

Both tibiae are exposed in anterior view (Fig 14D). They are gracile bones, approximately 50% longer than the femur. Elongated tibiae are present in other tapejarids (see Table 5), but not to

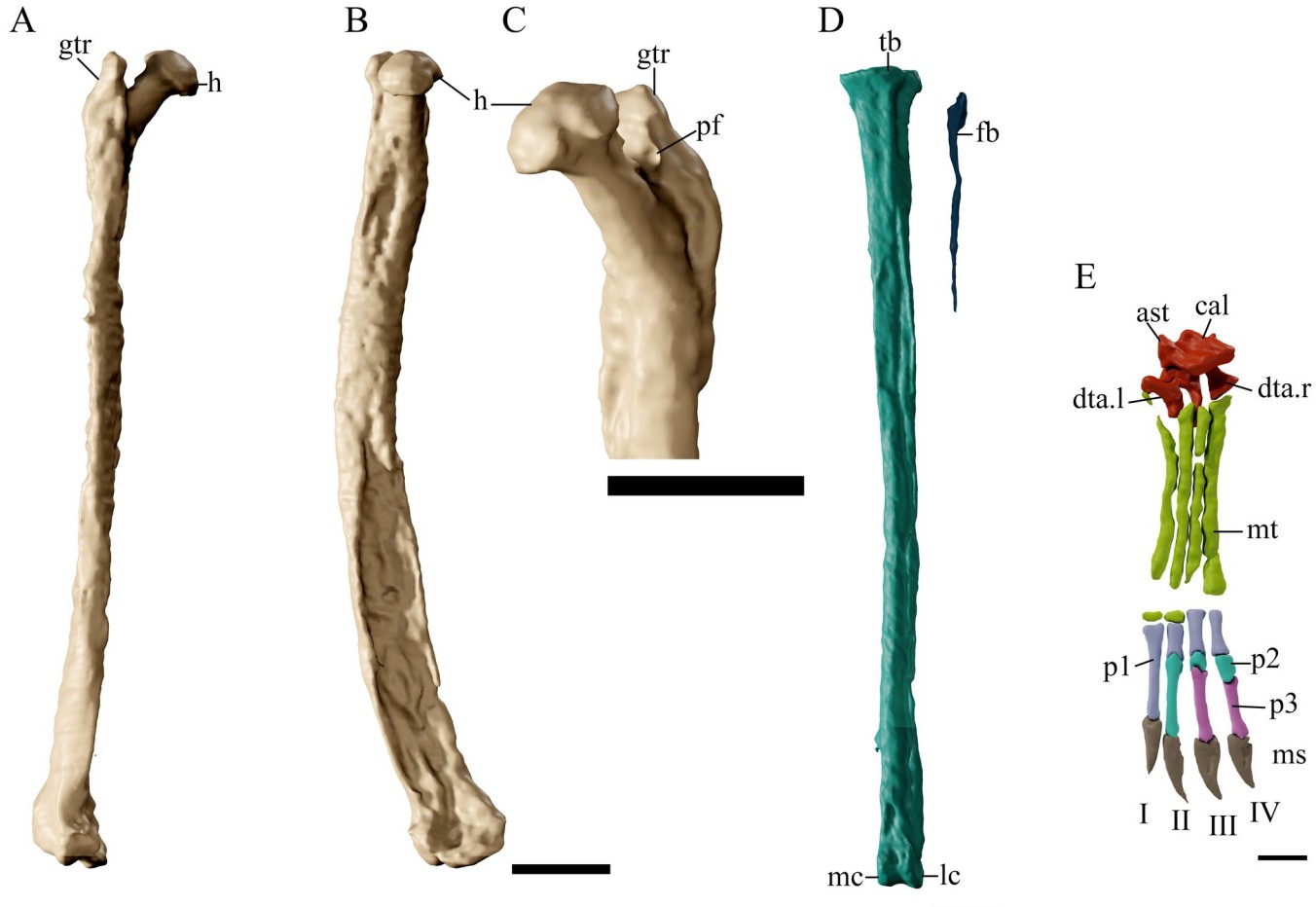

**Fig 14. *Tupandactylus navigans* GP/2E 9266 hindlimb elements.** 3D model of right femur in anterior view (A); medial view (B); proximal region in medio-posterior view (C); left tibia and fibula in anterior view (D); left pes (E). Abbreviations: ast, astragalus; cal, calcaneum; dta, distal tarsus; fb, fibula; gtr, great trochanter; h, femoral head; ms, medial sulcus; mt, metatarsus; p, phalanxes; pf, pneumatic foramen; tb, tibia. Scale bar = 20 mm (A-D); 10 mm (E).

the same degree as in dsungaripterids such as *'Phobetor' complicidens* (GIN125/1010; over 1.7) [52,53].

The tibia is proximally twice as broad as it is distally, with a very prominent proximal tubercle that is absent in thalassodromine tapejarids. The proximal articular surface is flattened as in other tapejarids, what differs from the rounded surface of ornithocheirids [47]. The left fibula is reduced to less than half the length of the tibia (0.39), a ratio just a little below that of other tapejarine tapejarids (e.g., 0.49 in *Ta. wellnhoferi*), but proportionally longer than in azhdarchids and dsungaripterids (0.16 in *Quetzalcoatlus* sp. and 0.22 in *Noripterus complicidens*). There is a small round expansion at the proximal end of the fibulae where two distal condyles are separated by a deep fossa.

## Tarsals and metatarsals

The proximal tarsals (Fig 14E) are formed by the calcaneum and astragalus. The former is flattened dorsoventrally, with an overall rectangular shape. The astragalus is concave posteriorly and convex anteriorly, a half-moon shape also seen in other tapejarids (such as *S. atavismus* and *Ta. wellnhoferi*; [6,7]). Two distal tarsals are present, with the left one longer (10.2 mm)

than the right one (6.2 mm). The left distal tarsal is rectangular shaped, whereas the right distal tarsal is sub-triangular. All metatarsals are similar in length and width, as appears to be the case for *Ta. wellnhoferi* SMNK PAL 1137 [6], but different from the condition in *S. dongi* and *S. lingyuanensis*, both whose metatarsals 3 and 4 are shorter than 1 and 2 [7,28]. The right metatarsals preserve their distal portions, with the fourth metatarsal being relatively shorter than the first three.

### Pes

As in all later-diverging pterodactyloids, there are only four pedal digits. The phalanges are thin and elongated, with phalangeal a formula of "2-3-3-4-0" (Fig 13E). The pedal ungual phalanges are shorter (10.7–14.1 mm long) than manual (24.5–27.3 mm long), but present the same mediolateral sulcus, broad and rounded flexor tubercles, and a distal ventral curvature.

### Phylogenetic analysis

The first phylogenetic analysis (without MN 6588-V) recovered 132 MTPs of 371 steps each, CI = 0.613, and RI = 0.867 (Fig 15). As would be expected, the results recover the same topology as in [15]. *Tupa. navigans* was recovered as a member of the clade Tapejarinae, within a polytomy that also includes *Ta. wellnhoferi*, *Tupa. imperator*, *E. olcadesorum*, and *C. dobruskii*. This clade is supported by one unambiguous synapomorphy [deep and broad ossified dentary sagittal crest (84)]. *Tupandactylus navigans* possesses two autapomorphies [presence of a notarium (113); humeral length less than 80% of femoral length (127)]. The scoring of GP/2E 9266 does not differ from that of the holotype of *Tupa. navigans* SMNK PAL 2344, but adds new anatomical data for the taxon. It is worth mentioning that the scoring of *Tupa. imperator* also does not differ from that of *Tupa. navigans*. The second phylogenetic analysis included the specimen MN 6588-V as a separate OTU and recovered 469 MTPs of 373 steps each. This analysis recovers the same topology as the first analysis, with MN 6588-V found as a representative of the Tapejarinae, in a polytomy with the abovementioned tapejarines.

## Discussion

### New information on *Tupandactylus navigans* anatomy

Specimen GP/2E 9266 (Figs 16 and 17) is the most complete Brazilian tapejarid described thus far. Although *Ta. wellnhoferi* and *C. dobruskii* are known from several individuals, no single specimen preserved the skeleton to the same degree as GP/2E 9266. The new material presents all the diagnostic characteristics of *Tupa. navigans*, namely a striated premaxillary crest, a perpendicular supra premaxillary bony process, and a short parietal crest [9]. In addition, we identified previously unrecorded diagnostic features that distinguish *Tupa. navigans* from other tapejarine pterosaurs, such as a deep, blade-shaped dentary crest (dentary crest height/ mandibular ramus height (DCH/MRH) ratio = 5.3) with a subvertical posterior margin, and cervical postzygapophyses displaying lateral longitudinal grooves. This is also the first time that some features are reported for Tapejarinae, such as the fusion of the atlas-axis complex, the presence of a notarium and a synsacral supraneural plate (absent in *Vectidraco daisymorrisae*), and metacarpals I-III barely reaching the proximal half of metacarpal IV. The presence of both dorsal and ventral foramina in the humeri of GP/2E 9266 is also worth noting. This feature was described as autapomorphic for *Ta. wellnhoferi* [6], but may be a synapomorphy of Tapejarinae.

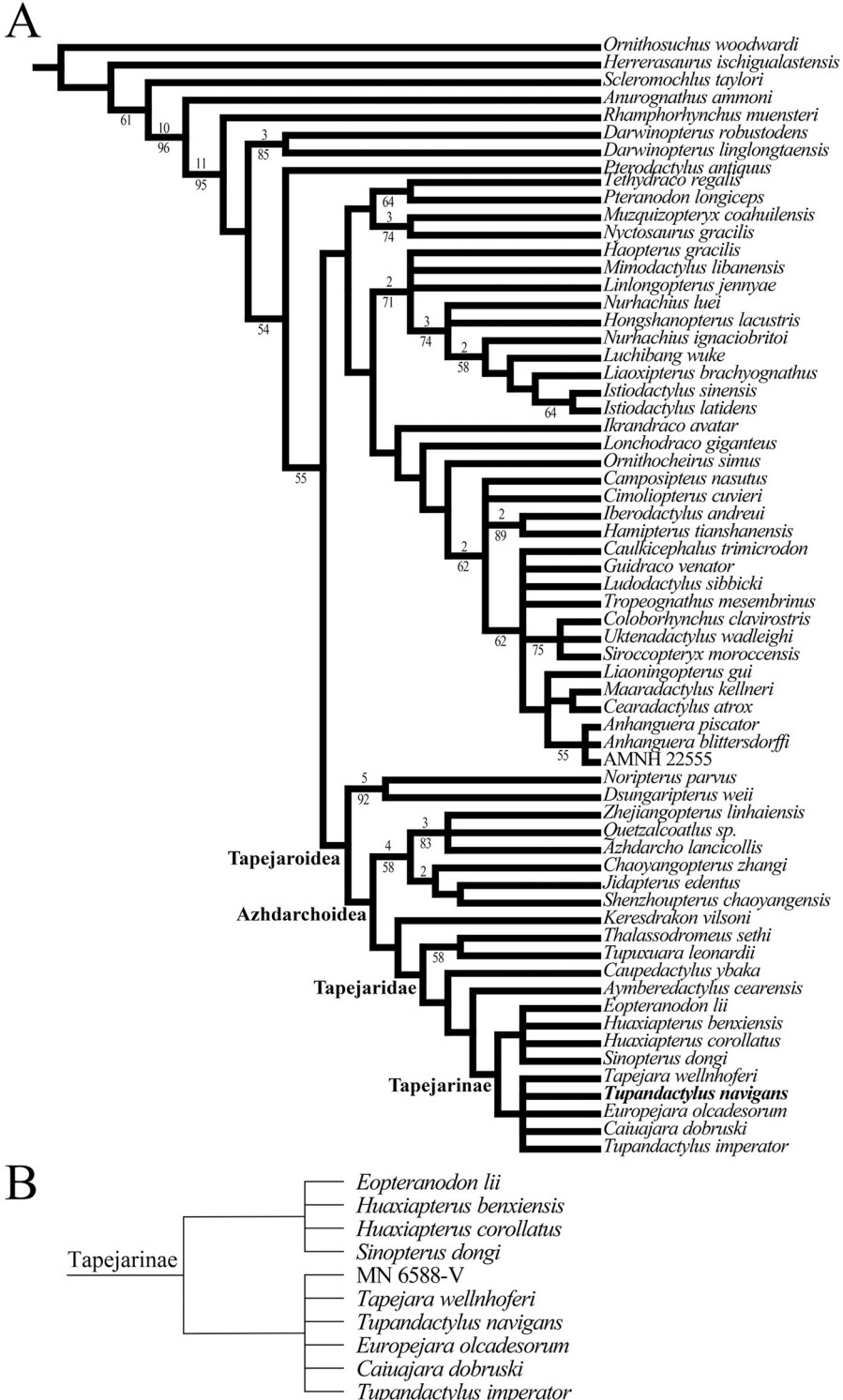

**Fig 15. Phylogenetic analysis using the character matrix of [14].** Strict-consensus tree recovered from the first phylogenetic analysis (A) and including specimen MN 6588-V, showing only Tapejaridae clade (B). Bremer support (> 1) is found over the nodes, and bootstrap (> 50) under the nodes.

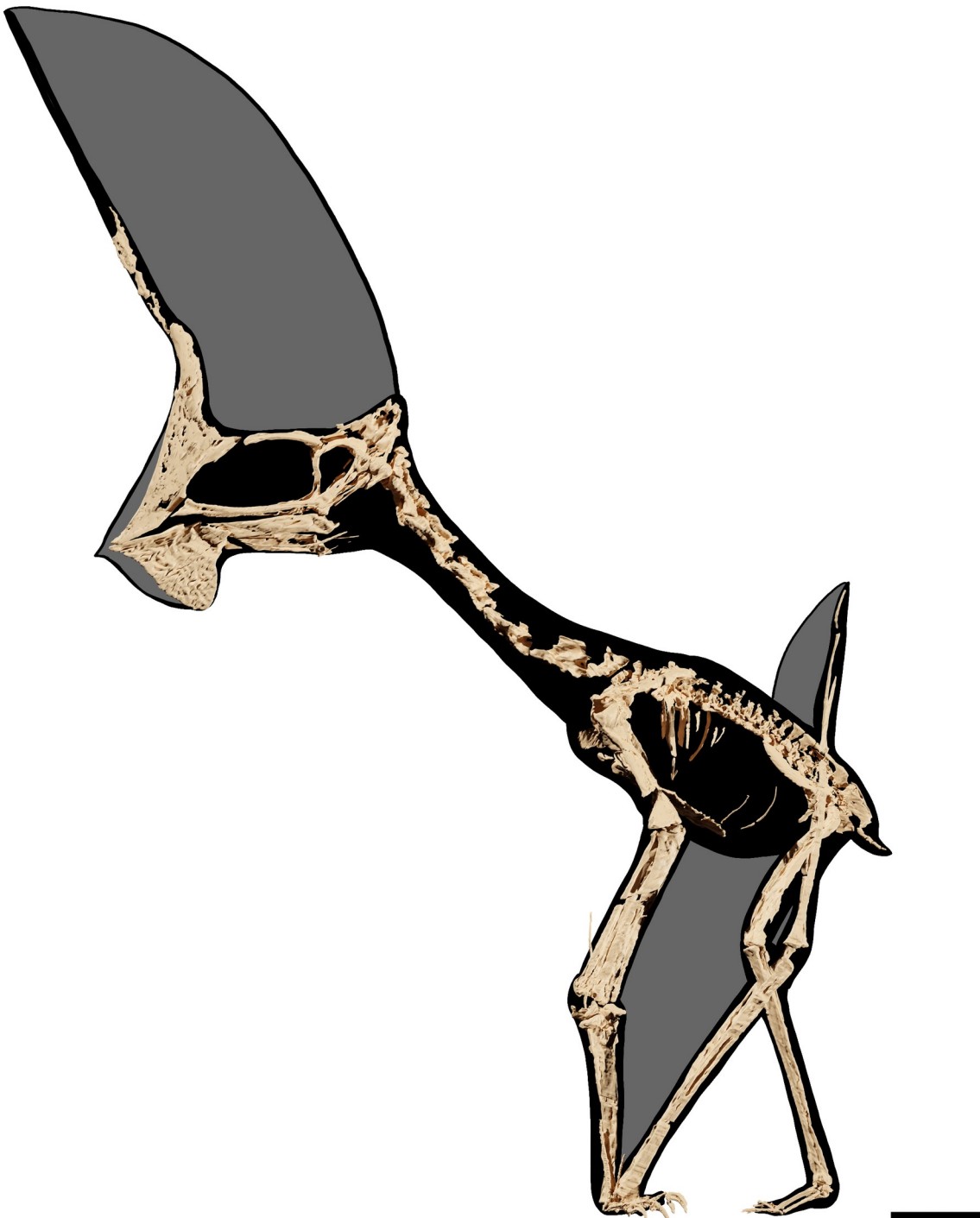

**Fig 16. Skeletal mount of *Tupandactylus navigans* specimen GP/2E 9266.** Skeletal mount of all bony elements recovered with the image segmentation of specimen GP/2E 9266. Scale bar = 50 mm.

## Ontogeny of specimen GP/2E 9266

The fusion of the premaxillomaxilla is reported even for immature pterodactyloid specimens, demonstrating that it occurs early in ontogeny [44]. Some other osteological features, however,

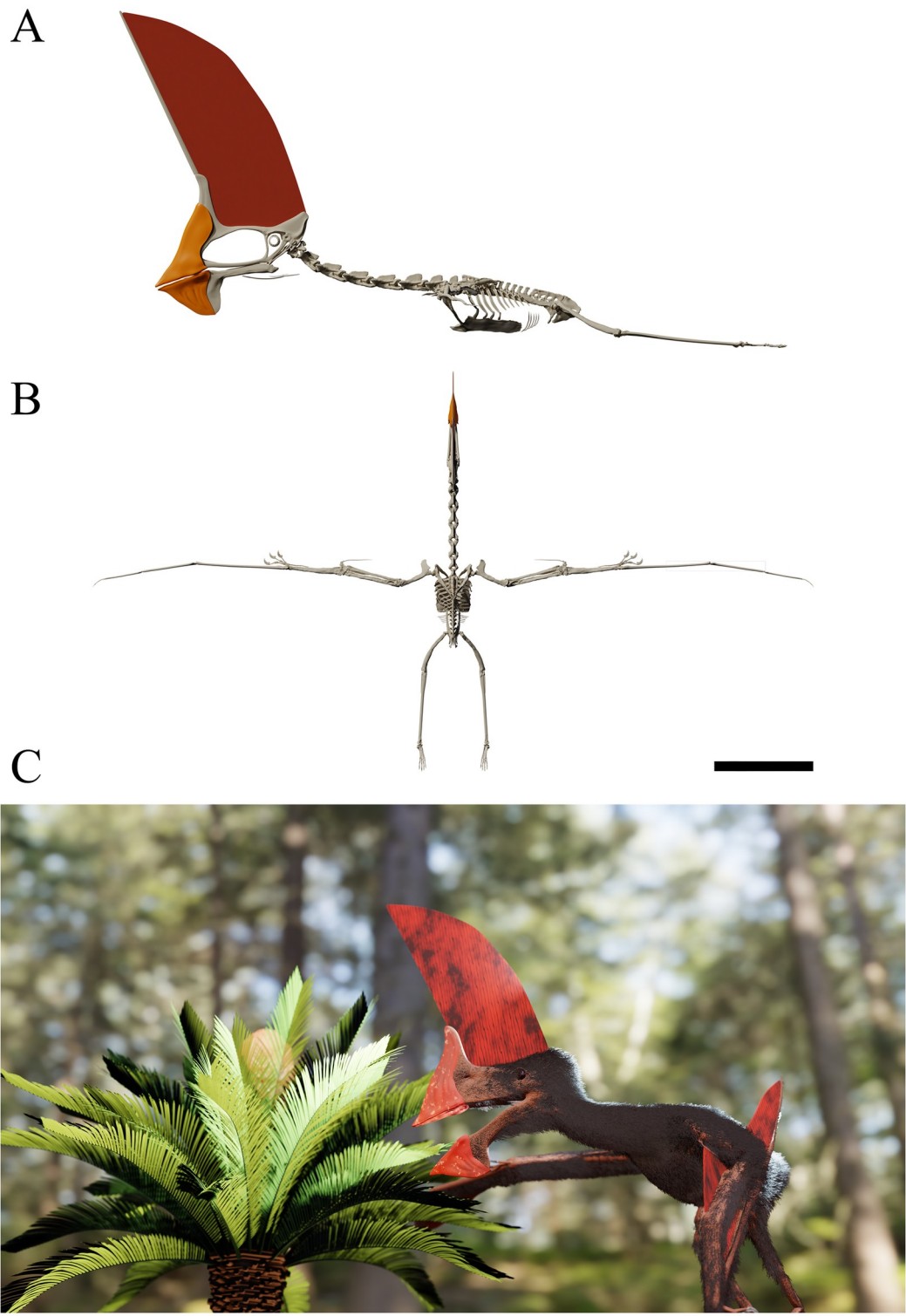

**Fig 17. Skeletal and living reconstruction of *Tupandactylus navigans* based on specimen GP/2E 9266.** Skeletal reconstruction of *Tupandactylus navigans* based on specimen GP/2E 9266 in lateral (A) and dorsal (B) views. Living interpretative reconstruction of *Tupa. navigans* based on specimen GP/2E 9266 (C). Scale bar = 50 mm.

apparently suggest an advanced ontogenetic stage for GP/2E 9266, such as the fusion of the extensor tendon process [65,69,75,76] and anterior dorsal vertebrae fused into a notarium. Notarium development of the new specimen fits state SN3 of [55]. According to these authors, stage SN3 occurs when all notarial centra are fused, with the first notarial neural spines also fused. In GP/2E 9266, the unfused tibiotarsus is the single skeletal feature arguing for a not fully mature ontogenetic stage [77]. Yet, as tapejarid ontogeny is not fully understood, it is still unclear how *Tupa. navigans* fits ontogenetic models based on other pterosaur taxa. As such, the integration of all those proxies for individual maturity indicates that GP/2E 9266 was a mature individual at the time of death.

### *Tupandactylus navigans* vs. *Tupa. imperator*

When first described, *Tupa. navigans* was differentiated from *Tupa. imperator* based on the perpendicularly oriented sagittal crest and the absence of a posterior expansion of the parietal crest [9]. The new material furthers differentiates the two species, based on traits that cannot be explained by taphonomy or ontogeny. Indeed, specimens from both species show similar preservation and presumably represent mature individuals, as suggested here for GP/2E 9266 and by [8,10] for the holotype (MCT 1622-R) and specimen CPCA 3590 of *Tupa. imperator* (Fig 18). Hence, we show that *Tupa. navigans* and *Tupa. imperator* also differ in characters

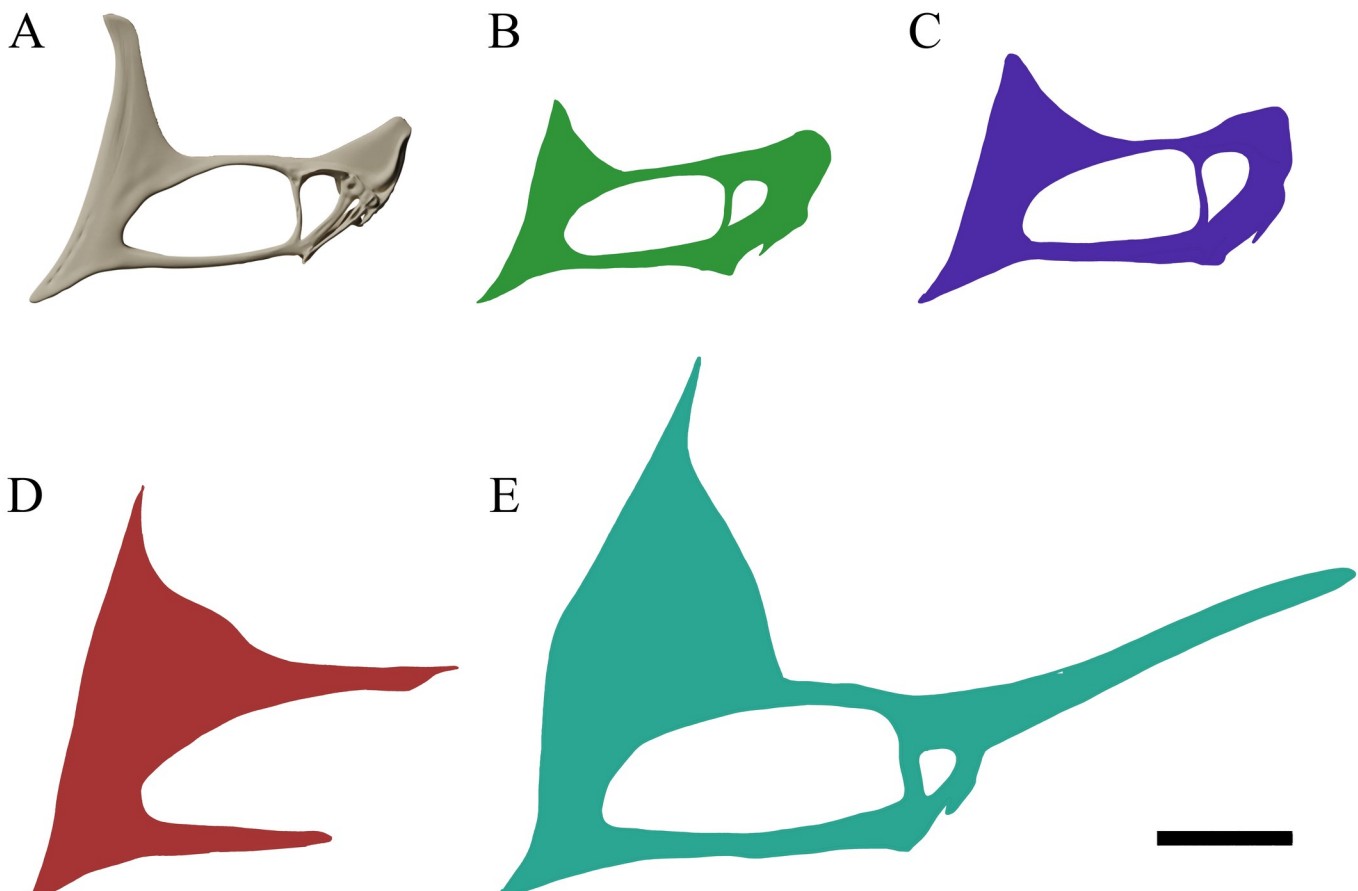

**Fig 18. Comparison of the skulls of *Tupa. navigans* and *Tupa. imperator*.** Comparison of the skulls of *Tupa. navigans* GP/2E 9266 (A), *Tupa. navigans* SMNK PAL 2344 (B), *Tupa. navigans* SMNK PAL 2343 (C), *Tupa. imperator* CPCA 3590 (D), and *Tupa. imperator* MCT 1622-R (E). Scale bar = 100 mm.

such as a proportionally larger dentary crest in *Tupa. navigans*. Additionally, the dentary crest of *Tupa. imperator* (CPCA 3590) is steeper anteriorly, whereas that of *Tupa. navigans* (GP/2E 9266) has a steeper posterior margin. Known specimens of *Tupa. navigans* are smaller than those of *Tupa. imperator*, and the skull proportions vary between both taxa (length-height ratio of 3.2 in GP/2E 9266 and of 3.6 for *Tupa. imperator*). However, the length-height ratio of the skull is highly variable between specimens, as can be observed for the holotype of *Tupa. navigans* SMNK PAL 2344, the skull of which is relatively longer than that of GP/2E 9266, with a length-height ratio of 2.5 (as noted in [10]). Hence, the skull length-height ratio seems not to be a strong basis to differentiate *Tupa. navigans* and *Tupa. imperator*.

The abovementioned differences do not rule-out the possibility that sexual dimorphism is the real explanation for the separation of both taxa. Both the sagittal and dentary crests might have worked as mating displays, what is arguable for pterosaur species with strong allometric growth or definite crest-related sexual dimorphism (e.g., [9,75,78,79]). Therefore, *Tupa. navigans* and *Tupa. imperator* could indeed represent two morphotypes of a single, sexually dimorphic species, and mutual sexual selection is not discarded [80]. Testing this hypothesis is, however, beyond the scope of the present work, and may depend on more detailed descriptive work for both species.

## Non-ossified tissue

A peculiarity of the rhamphotheca in GP/2E 9266 is the gap formed between the premaxillary and dentary (Fig 19). This gap prevents total occlusion of *Tupa. navigans* jaws, and could be related to a particular feeding strategy, such as the hard-plant diet inferred for tapejarids [4]. The presence of a presumably keratinous rhamphotheca covering the anterior rostrum of edentulous pterosaurs was already reported for tapejarid pterosaurs, such as for *Tupa. imperator* (CPCA 3590) and *Tupa. navigans* (holotype SMNK PAL 2344) [9,10]. Analogous to that of birds, this structure has been associated with energy storage and absorption of transmitted loads to the bone during the bite, what may be related to the lack of teeth [81]. In pterosaurs such as *Tupa. navigans*, the rhamphotheca could have been a hook-like structure operating to catch or to manipulate small food items [81].

The soft-tissue sagittal crest of GP/2E 9266 (Figs 5 and 19) is almost completely preserved, with only the apical region either missing or embedded in the sediment. Therefore, GP/2E 9266 possesses the most complete soft-tissue sagittal crest among tapejarines. The convex posterior margin of the crest displayed by the new specimen was suggested to be present also in *Tupa. imperator* (CPCA 3590, [10]) and in the holotype of *Tupa. navigans* (SMNK PAL 2344 [9]). As in CPCA 3590, differential color patterns in the soft-tissue crest may be related to oxidation. A thorough study of soft-tissue preservation in GP/2E 9266, applying UV-light and SEM, is underway.

## MN 6588-V as *Tupandactylus*

Sayão & Kellner [19] assigned an articulated axial series, including a notarium, and girdles (specimen MN 6588-V) to Tapejaridae mainly based on the presence of a tuberculum at the ventroposterior margin of the coracoid. The overall morphology of the axial series of MN 6588-V closely resembles that of GP/2E 9266 in both proportion and shape. MN 6588-V differs from GP/2E 9266, however, in two main aspects: the number of fused notarial vertebrae and the number of vertebrae in both dorsal and sacral series. MN 6588-V notarium is composed by the first four dorsal vertebrae, whereas in GP/2E 9266 it is composed of five elements. MN 6588-V dorsal and sacral vertebral counts are 11 and 7, respectively, against 10 and 5 in GP/2E 9266. The axial series is also relatively longer in MN 6588-V (dorsal series length = 178.5 mm,

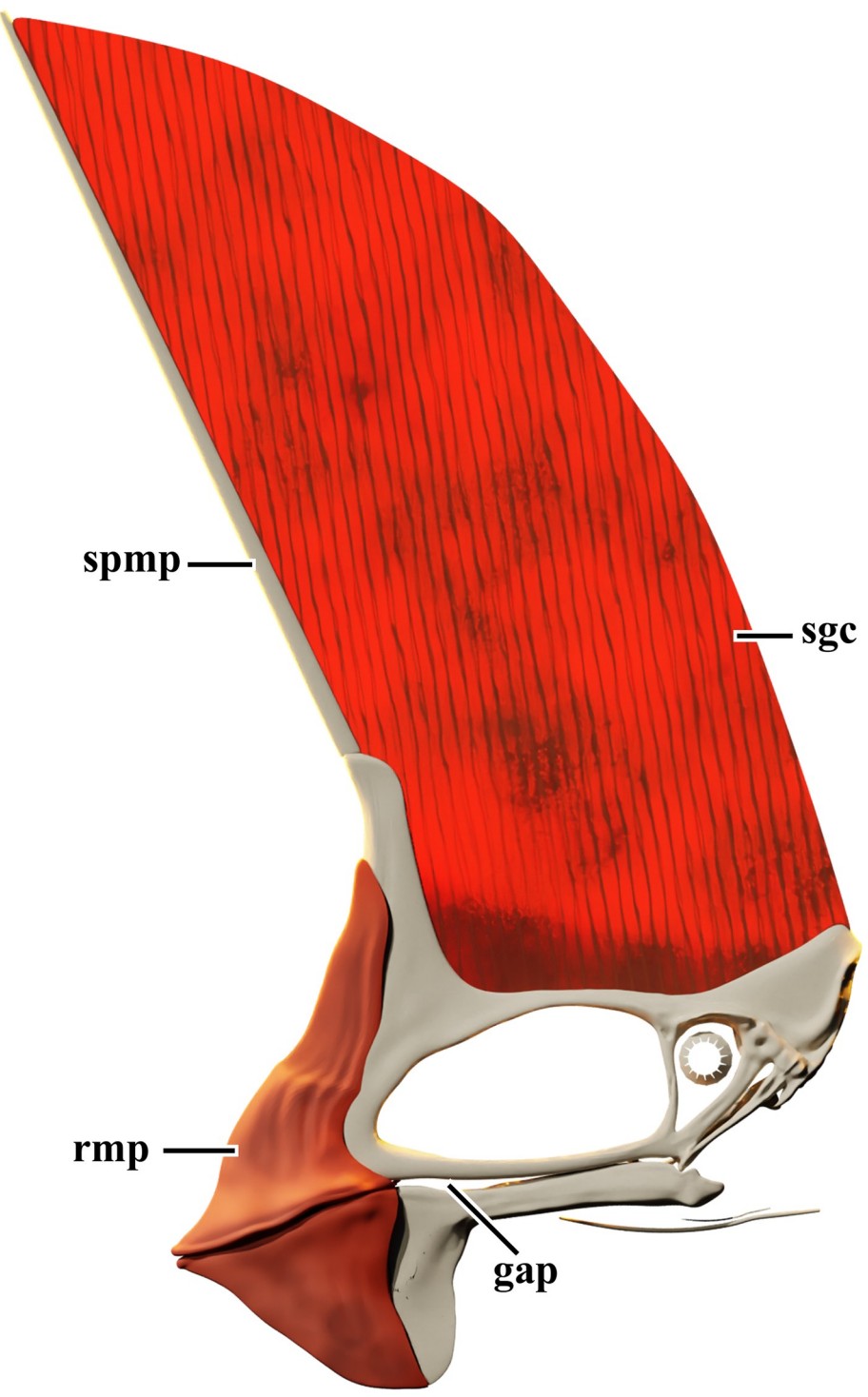

**Fig 19. Skull reconstruction of *Tupandactylus navigans* specimen GP/2E 9266.** Skull reconstruction of *Tupandactylus navigans* based on specimen GP/2E 9266 showing the non-ossified tissue and the jaw gap. Scale bar = 100 mm.

against 134.2 in GP/2E 9266). These differences, however, can be related to intraspecific variation [45], as no further diagnostic feature separates MN 6588-V from GP/2E 9266. Yet, both differ from other tapejarines by the presence of the notarium (as abovementioned). Due to the differences between MN 6588-V and GP/2E 9266 and the gap in our understanding of *Tupa. navigans* intraspecific variation, the specimen MN 6588-V cannot be confidently assigned to *Tupa. navigans*. However due to the abovementioned similarities to the specimen GP/2E 9266, MN 6588-V is here regarded as *Tupandactylus* sp.

## Implications to flight

The presence of a notarium in GP/2E 9266 is a novel feature for tapejarines, which opens debate to new possibilities regarding flight and terrestrial capabilities of *Tupa. navigans*. This structure was not observed in *C. dobruskii*, a pterosaur species from which a presumably complete ontogenetic series is known. It is worth mentioning that the wingspan estimate for *C. dobruskii* is close to that observed in GP/2E 9266, so that the absence of a notarium is not explained by the difference in sizes. The fusion of the first dorsal vertebrae into a notarium creates a stout structure that increased bending and torsion resistance, helping pterosaur in active flight [57]. Also, the deltopectoral crest of GP/2E 9266 forms a broad surface for the anchorage muscles such as the large pectoralis, a primary wing depressor whose division into caudal and cranial branches may have had an important role during gliding [82]. Moreover, the long fourth metacarpal and slender hindlimbs would have made a standing quadrupedal takeoff feasible [57,83]. Ultimately, the first wing phalanx length compared to metacarpal IV length in GP/2E 9266 is comparatively short (first wing phalanx length to metacarpal IV length in GP/2E 9266 = 0.58. For anhanguerids, this ratio is closer to 0.4 [64]). As such, the characteristic forelimb hypertrophy of specialized pterodactyloids is only moderately present in *Tupa. navigans* [43,64,84]. The relatively longer forelimbs and the long cervical series may argue for a terrestrial foraging lifestyle, although not the terrestrial-stalking ecomorphology proposed by Witton and Naish [72,85]. In fact, the cranial morphology of *Tupa. navigans* closely resembles that of *Ta. wellnhoferi*, the diet of which was suggested to include hard-plant material [4]. Although the crests in pterosaurs were possibly used as social and/or sexual features (see [80]), Frey and colleagues [9] argued that, in order to have an aerodynamically functional crest, *Tupa. navigans* should have had a short neck or tendon locks on its cervical vertebrae. Neither of these are observed in GP/2E 9266, in which the cervical series comprises over 55% of the total axial length (317 mm x 564 mm of the total length), and no ossified tendons are visible in pre-notarial vertebrae. This could indicate that the aberrant crest may have restricted *Tupa. navigans* to short-distance flights, such as to flee from predators. However, the biomechanical meaning of the set of features seen in GP/2E 9266 regarding its flight and/or terrestrial capabilities requires further studies, which is beyond the scope of this paper.

## Conclusion

The specimen here described is the most complete articulated tapejarid skeleton thus far recovered in Brazil. The premaxillary crest shape, dentary crest proportions and shape, as well as axial and appendicular skeletal anatomy (including the presence of a notarium) are novel features observed in GP/2E 9266, which allowed to propose an emended diagnosis for *Tupa. navigans*. Differences between *Tupa. navigans* and *Tupa. imperator* may be due to sexual dimorphism, but this requires further investigation. The specimen MN 6588-V is regarded here as *Tupandactylus* sp., due to its similarities to the post-cranium of GP/2E 9266. The flight capabilities and potential terrestrial foraging lifestyle was briefly discussed, but further requires

further biomechanical analysis. The new specimen considerably improves our understanding of tapejarid anatomy, taxonomy, and ecomorphology.

## Supporting information

**S1 File.**
(TNT)

**S2 File.**
(TNT)

## Acknowledgments

The authors would like to thank Ivone Cardoso Gonzales and Juliana de Moraes Leme (IGc/USP) for the access of the material. We want to thank Beatriz Verdasca Aceto (MAE/USP) and Antonio Carlos Matioli (Hospital Universitário/USP), for the tomography of the material. We also want to thank Alexander O. Averianov and David W. E. Hone for the revision and feedback that greatly improved this manuscript, and Anna Carolina Dias de Almeida (IB/USP) for photos of the material and overall help during the development of the present work. The authors would like to thank Willi Hennig Society for the gratuity of TNT software. VB want to thank Thermo Fisher Scientific for the student license to Avizo software. For allowing access to relevant fossil collections, FLP is indebted to Mark Norell and Carl Mehling (AMNH); Álamo Saraiva and João Kerensky (MPSC); Oliver Rauhut and Markus Moser (SNSB-BSPG); Alexander Kellner and Helder Silva (MN); Eberhard Frey (SMNK); Rainer Schoch (SMNS); Sandra Chapman and Lorna Steel (NHMUK); Dan Pemberton and Matt Riley (CAMSM).

## Author Contributions

**Conceptualization:** Victor Beccari, Luiz Eduardo Anelli, Fabiana Rodrigues Costa.

**Data curation:** Victor Beccari.

**Formal analysis:** Victor Beccari.

**Investigation:** Victor Beccari, Felipe Lima Pinheiro, Luiz Eduardo Anelli, Octávio Mateus, Fabiana Rodrigues Costa.

**Methodology:** Victor Beccari, Ivan Nunes, Luiz Eduardo Anelli, Fabiana Rodrigues Costa.

**Project administration:** Victor Beccari, Fabiana Rodrigues Costa.

**Software:** Victor Beccari.

**Visualization:** Victor Beccari.

**Writing – original draft:** Victor Beccari, Felipe Lima Pinheiro, Ivan Nunes, Luiz Eduardo Anelli, Octávio Mateus, Fabiana Rodrigues Costa.

**Writing – review & editing:** Victor Beccari, Felipe Lima Pinheiro, Ivan Nunes, Luiz Eduardo Anelli, Octávio Mateus, Fabiana Rodrigues Costa.

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
