## [Decision Letter · Decision Letter 0]

13 Apr 2021

PONE-D-20-38270

Osteology of an exceptionally well-preserved tapejarid skeleton from Brazil: revealing the anatomy of a curious pterodactyloid clade

PLOS ONE

Dear Mr. Beccari,

Thanks for choosing PLOS ONE as the venue to publish your research!

We have now secured two revisions for your manuscript. Sorry for the delay, but it was hard to find suitable reviewers along the passage of the year.

Both reviewers are generally happy with the MS, so I ask you to consider all the modifications they suggest (accept or reply to them accordingly) and proceed so submit a revised version of the manuscript.

We look forward to receiving your revised manuscript.

Kind regards,

Max Cardoso Langer

Academic Editor

PLOS ONE

Journal Requirements:

We note that you have stated that you will provide repository information for your data at acceptance. Should your manuscript be accepted for publication, we will hold it until you provide the relevant accession numbers or DOIs necessary to access your data. If you wish to make changes to your Data Availability statement, please describe these changes in your cover letter and we will update your Data Availability statement to reflect the information you provide.

Please include captions for your Supporting Information files at the end of your manuscript, and update any in-text citations to match accordingly. Please see our Supporting Information guidelines for more information: http://journals.plos.org/plosone/s/supporting-information.

Reviewers' comments:

Reviewer's Responses to Questions

**Comments to the Author**

1. Is the manuscript technically sound, and do the data support the conclusions?

Reviewer #1: Yes

Reviewer #2: Yes

2. Has the statistical analysis been performed appropriately and rigorously? 

Reviewer #1: N/A

Reviewer #2: N/A

3. Have the authors made all data underlying the findings in their manuscript fully available?

Reviewer #1: Yes

Reviewer #2: Yes

4. Is the manuscript presented in an intelligible fashion and written in standard English?

Reviewer #1: No

Reviewer #2: Yes

5. Review Comments to the Author

Reviewer #1: Review of “Osteology of an exceptionally well-preserved tapejarid skeleton from Brazil: revealing the anatomy of a curious pterodactyloid clade” by Victoer Beccari and colleagues.

The paper describes in detail anatomy of a remarkable complete and well preserved skeleton with remains of soft tissues of the tapejarid pterosaur Tupandactylus navigans from the Lower Cretaceous Crato Formation of Brazil. The paper is certainly an important contribution to our understanding of pterosaurs and should be published after minor revision. The paper is generally well written although English needs improvement. I have only few comments to the paper which are introduced in the attached pdf file of the paper.

Alexander Averianov

Reviewer #2: See the attached marked-up document for full details and comments. Fundamentally the paper is fine but there are a few areas that I think can be improved and would make it much stronger.

Needs comparisons to other specimens of this species.

An interpretative drawing of the skull here and in comparison to other Tupandactylus species would be very useful.

Why not code multiple specimens of the species into the analysis?

The ontogeny section needs to be cleared up.

The ecological section is poorly referenced and contains many unsupported ideas.

6. PLOS authors have the option to publish the peer review history of their article (what does this mean?). If published, this will include your full peer review and any attached files.

Reviewer #1: **Yes: **Alexander Averianov

Reviewer #2: **Yes: **David Hone

---

## [Author Response · Author response to Decision Letter 0]

31 May 2021

For Reviewer 1 - Alexander Averianov

All inquiries regarding the proper writing were accepted.

The reviewer comments and authors replies are found in the "Response to Reviewers" document.

All the comments were taken into consideration and the suggested changes were accepted.

For reviewer 2 - David Hone

All inquiries regarding the proper writing were accepted.

The reviewer comments and authors replies are found in the "Response to Reviewers" document. 

General issues:

Needs comparisons to other specimens of this species.

R: accepted and added a section, a table (Table 5) and a figure (Fig. 18) to compare GP/2E 9266 to the described specimens of T. navigans and T. imperator.

An interpretative drawing of the skull here and in comparison to other Tupandactylus species would be very useful.

R: accepted and figure added.

Why not code multiple specimens of the species into the analysis? 

R: accepted. The holotype of T. navigans shows no differences in coding (section added to the text), and specimen MN 6588-V (a postcranial material assigned to tapejarid by Sayão and Kellner, 2006) was codded in the phylogeny and added to the discussion.

The ontogeny section needs to be cleared up.

R: accepted and this section was revised.

The ecological section is poorly referenced and contains many unsupported ideas.

R: The ecological section was rewritten and better structured.

---

## [Editor Report · Decision Letter 1]

22 Jun 2021

PONE-D-20-38270R1

Osteology of an exceptionally well-preserved tapejarid skeleton from Brazil: revealing the anatomy of a curious pterodactyloid clade

PLOS ONE

Dear Dr. Beccari,

Thank you for submitting your manuscript to PLOS ONE. I have now evaluated its revised version on the light of the suggestions provided by the two reviewers. I understand that you gave adequate answer to most of their comments, so that we not need to go through a second round of revisions.

Nevertheless, I would like to add some editorial suggestion to the text. This is attached using the modified MS file you provided, in which I have accepted all of your modifications and further included (using track-changes) my suggestions. Please, either accept them or provide a justification for not doing so.

There are also some other minor aspects I would like to highlight, see "Additional Editor Comments" below.

We look forward to receiving your revised manuscript.

Kind regards,

Max Cardoso Langer

Academic Editor

PLOS ONE

Journal Requirements:

Additional Editor Comments:

1 - Abbreviation of generic names and citation of species:

Please, always provide the full species name (not only the generic epithet) if this is known (otherwise use “sp.”). As for the abbreviation of the generic epithet, this is fine and welcome, but has to be patronised. So, use a full-word epithet in the first time the species is mentioned in the abstract and in the main text; after that, always use the abbreviation (unless it is the first work of a sentence). As for generic epithets starting with the same set of letters, the abbreviation has to include a minimal number of letters enough to differentiate them (e.g., Tupu. and Tupa. for Tupuxuara and Tupandactylus). I attempted to change all such cases in the MS, but please give a second look.

2 – Phylogenetic analysis:

This needs additional information for clarity and, depending on the decision of the authors, also some implementations of the analysis itself.

You wrote, “The phylogenetic position of Tupa. navigans was accessed through the scoring of GP/2E 9266 using the character-taxon matrix of Hone et al. (2020)”. As far as I could follow, this was based on Kellner et al. (2019), which was based on Holgado et al. (2019), which was based on Vullo et al. (2012), which was based on many sources. What is important to mention is [1] on what basis was Tupa. navigans scored on those previous matrices (which specimens, by which authors), and [2] if you just complemented their codification with new data from GP/2E 9266 or if you employed a completely de novo codification based only on GP/2E 9266. It seems that the latter was the case, if so, you have to clearly state if your codification is totally congruent with the previous codification of Tupa. navigans (i.e., no character was scored differently between yours and previous codifications). If this is not the case, you should consider including GP/2E 9266 as an OUT independent from Tupa. navigans (as previously scored) in those matrices; show and discuss the results. I see that this is briefly discussed in the “Phylogenetic Analysis” (apparently saying that the codifications are the same), but this needs to be clearly stated in the “M&M”, beforehand.

You also wrote, “The second phylogenetic analysis included the specimen MN 6588-V as a separate OTU”. Firstly, you have to mention this second analysis in the “M&M” section. Secondly, it is unclear from the sentence above if MN 6588-V was previously included in the data sets as part of a given OUT or if the specimen was never considered in those data sets. You have to make this point clear.

Finally, you mention that the tree in Fig. 15B is a “reduced strict-consensus tree”. You have to explain how this reduced consensus tree was built in the “M&M” and then provide more detailed results, e.g., which taxa were excluded and how.

3 – Tables:

Please, do not provide them within the main text, but each in a separate file formatted and labelled accordingly.

4 – You wrote, “MN 6588-V can be regarded as Tupandactylus sp. due to the similarities to the postcranial material of GP/2E 9266”. Question: why not to Tupa. navigans, considering that GP/2E 9266 belongs to that species? In order to state that it belongs to the genus, but not to either of its species, you would have to say (if this is the case) that it does not preserve the putative differences between both species. If this is not the case, perhaps you can define the species of MN 6588-V. In any case, further explanation are needed here. See also if this leads to modifications in the conclusion.

5 – Comments by Dr. David Hone

Three comments by reviewer Dr. David Hone were not clearly tacked in the modified MS. Please respond to these queries: [1] about the hook-like structure of the rhamphotheca, Dr. Hone asked “I don't understand this at all. It doesn't look like it is hooked and I do not know how a 'peg' would operate. Please explain.”; [2] In the sentence “wing phalanges length compared to the inner-wing length in GP/2E 9266 is comparatively short” Dr. Hone asked for companions; [3] About the sentence “Frey and colleagues argued that, in order to have an aerodynamically functional crest … ”, Dr. Hone also commented that “they assume crests were functional which is very unlikely (see Hone et al. 2010). If crests were functional we'd expect them to converge on a very small number of forms, instead they show very high diversity which is a signature of socio-sexually structures”. Please respond to his query.

Max

---

## [Author Response · Author response to Decision Letter 1]

28 Jun 2021

All the editors and reviewers inquiries were taken into consideration and properly implemented in the Manuscript. The inquiries have been accepted, with the exception of point 3 - Tables, as the Journal Guidelines demanded the tables within the text. We thank the editor for his revision.

---

## [Editor Report · Decision Letter 2]

5 Jul 2021

Osteology of an exceptionally well-preserved tapejarid skeleton from Brazil: revealing the anatomy of a curious pterodactyloid clade

PONE-D-20-38270R2

Dear Dr. Beccari,

We’re pleased to inform you that your manuscript has been judged scientifically suitable for publication and will be formally accepted for publication once it meets all outstanding technical requirements.

At this time, please also include the three minor changes requested by the editor (see below)

Kind regards,

Max Cardoso Langer

Academic Editor

PLOS ONE

Additional Editor Comments:

Only three minor changes:

Line 110: change “differed GP/2E 9266 scoring” for “differed from the scoring of GP/2E 9266”

Line 122: delete “The reduced tree consists of the clade Tapejarinae.” (see below)

Line 897: change “Strict-consensus tree recovered from the phylogenetic analysis (A) and reduced strict consensus tree recovered including specimen MN 6588-V (B)” for “Strict-consensus tree recovered from the first phylogenetic analysis (A) and including specimen MN 6588-V, showing only Tapejaridae clade (B)”. Otherwise, it gives the impression that you built a reduced-consensus tree.

---

## [Editor Report · Acceptance letter]

15 Jul 2021

PONE-D-20-38270R2 

Osteology of an exceptionally well-preserved tapejarid skeleton from Brazil: revealing the anatomy of a curious pterodactyloid clade 

Dear Dr. Beccari:

I'm pleased to inform you that your manuscript has been deemed suitable for publication in PLOS ONE. Congratulations! Your manuscript is now with our production department. 

Kind regards, 

on behalf of

Dr. Max Cardoso Langer 

Academic Editor

PLOS ONE